# Implicit Bias and Invariance: How Hopfield Networks Efficiently Learn Graph Orbits

## Abstract

Many learning problems involve symmetries, and while invariance can be built into neural architectures, it can also emerge implicitly when training on group-structured data. We study this phenomenon in classical Hopfield networks and illustrate how they can infer the isomorphism class of a graph from a small, random sample. Our results reveal that: (i) graph isomorphism classes can be represented within a three-dimensional invariant subspace, (ii) using gradient descent to minimize energy flow (MEF) has an implicit bias toward norm-efficient solutions, which underpins a polynomial sample complexity bound for learning isomorphism classes, and (iii) across multiple learning rules, parameters converge toward the invariant subspace as sample sizes grow. Together, these findings highlight a unifying mechanism for generalization in Hopfield networks: a bias toward norm efficiency in learning drives the emergence of approximate invariance under group-structured data [1].

## 1 Introduction

Here, we analyze the emergence of invariance arising implicitly during training in Hopfield networks (HNs) (Hopfield, 1982), which represent arguably the simplest example of an Associative Memory. Building on classical ideas  Rosenblatt (1958); Willshaw et al. (1969); Amari (1972); Little (1974); Pastur & Figotin (1977), HNs are recurrent neural networks consisting of $n$ linear-threshold McCulloch–Pitts neurons McCulloch & Pitts (1943) that can store binary patterns as distributed memories in the form of fixed-point attractors of recurrent dynamics. In the literature, HNs are usually associated with a particular Hebbian learning scheme called the Outer-Product Rule, but for the purposes of this work we also consider other standard training methods. This setting is intentionally minimal so that we can focus on developing novel mathematical tools for understanding generalization in a classical architecture. As data symmetry is not made explicit in this model, any invariance must arise from the interplay of the group structure in the data and the implicit bias of the learning rule in question. More specifically, and inspired by Hillar & Tran (2018); Hillar et al. (2021), this paper studies whether or not standard learning rules and objectives, notably minimization of the energy flow (MEF) Hillar et al. (2012), a tractable convex loss, *can learn the isomorphism class of a graph from a small, random subset.* Our key findings are as follows.

1. **HNs can memorize any graph isomorphism class.** We characterize the subspace of parameters invariant to edge-adjacency–preserving permutations (of which graph isomorphisms are a subset) (Lemma 4.6) and observe that this subspace aligns well with the parameters of successfully trained models (Fig. 5). Moreover, for any graph we give an explicit construction within this space that memorizes it (Lemma 4.7) as well as its isomorphism class.

2. **Implicit bias towards norm efficient solutions.** We reparameterize the MEF objective and show that gradient descent on it is directionally biased towards the solution to a hard-margin support vector machine (HSVM) problem on an induced linear representation (Corollary 3.1).

---

[1]**Statement on the use of LLMs.** Large language models (LLMs) were used to assist with literature search, checking and refining the clarity of writing, high level ideation and planning as well as organizing related work.

3. **Polynomial sample complexity suffices for orbit generalization.** Suppose $D \subset \{0,1\}^n$ is strictly memorizable with min-norm parameter $\boldsymbol{\theta}^*$, $\|\boldsymbol{x}\|_0 \leq m$ for all $\boldsymbol{x} \in D$, and let $\mathcal{D}$ be a distribution supported on $D$. We prove $N = \tilde{\Omega}(n\|\boldsymbol{\theta}^*\|^2 m\epsilon^{-2})$ random samples suffices for both HSVM and MEF to memorize new samples with probability at least $1 - \epsilon$ (Theorem 3.2). These theoretical results corroborate the empirical "few-shot-to-orbit" phenomenon we observe (Figs. 1, 2, 6, 8, 9, 10). Moreover, specializing to isomorphism classes this result implies a polynomial sample complexity in the number of vertices $v$, supporting a conjecture in Hillar et al. (2021) for the case of cliques. In this case, the critical number of samples for generalization appears to coincide with when these networks solve the Hidden Clique Problem (HCP) Dekel et al. (2014); see Fig. 6 in Appendix C.3. These critical sample counts can also be used to differentiate between combinatorial classes of graphs (see Fig. 2).

4. **Emergence of invariance.** We observe that as the sample size $N$ grows the learned parameters concentrate near the invariant subspace (Figs. 3, 5, 6, 9). For a simplified average HSVM surrogate, we prove the sample solution converges to the invariant set at rate $\tilde{O}(v^2/\sqrt{N})$ (Lemma 4.10), where $v$ is the number of vertices of the graph.

## 1.1 Related work

**Capacity of Hopfield networks.** The capacity of a Hopfield network depends on the learning rule and the structure of the data. For dense, uncorrelated random patterns under Hebbian learning, the statistical–mechanics analysis of (Amit et al., 1985) gives the classic linear law: reliable retrieval up to approximately 0.138 patterns per neuron with subsequent refinements via replica methods (Gardner, 1988; Krauth & Mézard, 1989). Coding-theoretic analyses further show that, for Hebbian constructions and exact recovery of randomly chosen patterns, one typically cannot exceed $n/(4\ln n)$ (McEliece et al., 1987) memories. More generally, Cover's classical bound Cover (1965) restricts the capacity for exact storage of dense, random data to only $2n$. Nonetheless, superlinear capacity is achievable for certain structured datasets. For example, sparse data having few active neurons can yield an increase of capacity to nearly quadratic in $n$ (Tsodyks & Feigel'man, 1988; Amari, 1989). Additionally, robust exponential memory has been observed for particular examples of group structured data (Hillar & Tran, 2018; Hillar et al., 2021); in particular, for storing all $k$-clique graphs and their hypergraph analogues. Our work builds on the observations of (Hillar & Tran, 2018; Hillar et al., 2021) by proving that all graph isomorphism classes are memorizable.

**Modern Hopfield Networks.** A line of recent investigation has sought to increase the capacity and retrieval properties of Hopfield networks by changing the energy function. Dense Associative Memories (DAMs) replace the classical quadratic energy with higher-order polynomial interactions, resulting in a capacity that scales polynomially with neuron count (Krotov & Hopfield, 2016; Horn & Usher, 1988). Building on this, Modern Hopfield Networks (MHNs) introduced a log-sum-exp energy function that allows the capacity to grow exponentially (Ramsauer et al., 2021; Demircigil et al., 2017). Our work provides a complementary perspective to these advancements by showing classical HNs can achieve exponential capacity by capturing the symmetry of the data in its parameters. Preliminary experiments suggest that DAMs do not generalize well in this setting (see Fig. 7, Appendix C.4).

**Generalization beyond the training set for HNs.** Theoretical study of classical HNs primarily focuses on storage limits, basins of attraction, and noise robustness around memorized patterns, rather than sample–complexity guarantees for generalization to patterns outside the training set. Earlier analyses of concept generalization in classical HNs investigate when networks capture latent data regularities (Fontanari, 1990). More recently, Random-Features Hopfield Models (RFHMs) exhibit learning phase transitions and even retrieval of previously unseen examples (Negri et al., 2023; Kalaj et al., 2025). These results complement but are distinct from our own; in particular, they do not provide sample–complexity bounds or analyze the emergence of invariance induced by a symmetry present in the data. In addition, while these results primarily use techniques from statistical physics, here we leverage tools from statistical learning theory.

**Emergent invariance through data augmentation and feature averaging.** One perspective on data augmentation is as orbit averaging over a symmetry group; in particular, empirical risk minimization on augmented data is equivalent to averaging features or predictions along group orbits. This has been

shown to induce approximate invariance and reduces estimator variance (Chen et al., 2020). From a kernel perspective, augmentation decomposes into first-order invariant feature averaging plus a second-order variance regularizer (Dao et al., 2019). Enforcing invariance through this averaging method yields provable generalization benefits in the context of invariant kernel regression (Elesedy & Zaidi, 2021). Beyond static estimators, recent results show that with full group augmentation deep ensembles become equivariant in expectation at all training times in the infinite-width limit (Gerken & Kessel, 2024) and that the expected predictions of group-convolutional networks match those of data-augmented conventional networks throughout training (Misof et al., 2025). Note a formal comparison between group equivariant architectures and data augmentation, without resorting to the infinite-width limit, is provided in Nordenfors et al. (2024). Finally, in regard to signatures of feature learning in the context of group structured data, then under certain conditions if a neural network is invariant to a finite group its trained weights recover the Fourier transform on that group (Marchetti et al., 2024). While these results assume explicit invariance, either through architectural design or by averaging over the full group orbit, here we ask whether simple learning rules can implicitly recover approximate invariance from a small, random sample of elements.

**Implicit bias.** A large body of work shows that, even without explicit regularization, certain learning dynamics have a preference for particular solutions. In particular, for classification using the logistic loss, gradient descent drives the parameter norm to infinity while the parameter direction converges to the max–margin classifier (Soudry et al., 2018; Ji & Telgarsky, 2018; Nacson et al., 2019). Our work leverages these results in order to show that standard learning rules for HNs are implicitly biased towards learning invariant representations when trained on group data.

## 2 Preliminaries

**Notation:** we use capitalized boldface characters to denote matrices, bold lowercase characters to denote vectors and non-bold lowercase characters to denote scalar values. If $\boldsymbol{x} \in \mathbb{R}^n$ is a vector then $x_i$ denotes the $i$th entry of $\boldsymbol{x}$. If $\boldsymbol{X} \in \mathbb{R}^{N \times n}$ then $\boldsymbol{x}_i \in \mathbb{R}^n$ denotes the $i$th row of $\boldsymbol{X}$ and to access individual entries of $\boldsymbol{X}$ we use the notation $x_{ij}$ or $[\boldsymbol{X}]_{ij}$, whichever is clearer in context. Whether a matrix, vector or scalar is deterministic or random is also inferred from context. We use $\Pi_a$ to denote the set of permutations on $[a]$ and $\mathcal{P}_a$ to denote the set of $a \times a$ permutation matrices. Overloading our notation, we also refer to $\mathcal{P}_a$ as the group of permutation matrices. Finally, if $\mathcal{H}$ is a group that acts on a set $\mathcal{A}$, then the orbit of $a \in \mathcal{A}$ under $\mathcal{H}$ is denoted $\mathrm{Orb}(a, \mathcal{H}) = \{ha \in A : h \in \mathcal{H}\}$.

**Associative Memory:** we consider a Hopfield network Hopfield (1982) with asynchronous dynamics but do not restrict ourselves to Hebbian learning. To this end, let $\mathrm{Sym}_0^n \subset \mathbb{R}^{n \times n}$ denote the set of symmetric, real, $n \times n$ matrices whose diagonal entries are zero, and let $\Theta = \mathrm{Sym}_0^n \times \mathbb{R}^n$. Clearly $\Theta$ is a convex vector space. We introduce the energy function $E : \{0, 1\}^n \times \boldsymbol{\Theta} \to \mathbb{R}$ defined as

$$E(\boldsymbol{x}; \boldsymbol{\theta}) = \frac{1}{2}\boldsymbol{x}^T \boldsymbol{W} \boldsymbol{x} + \boldsymbol{b}^T \boldsymbol{x}, \tag{1}$$

where $\boldsymbol{\theta} = (\boldsymbol{W}, \boldsymbol{b}) \in \boldsymbol{\Theta}$. Given an input binary vector $\boldsymbol{x} \in \{0, 1\}^n$, the Hopfield network generates a sequence of binary vectors $(\boldsymbol{x}(t))_{t \geq 0}$ through the following recurrence dynamics: with $\boldsymbol{x}(0) = \boldsymbol{x}$ then

$$x_j(t) = \begin{cases} \mathbb{1}(-\boldsymbol{w}_j^T \boldsymbol{x}(t-1) > b_j) & t \equiv j \pmod{n}, \\ x_j(t-1) & \text{otherwise} \end{cases} \tag{2}$$

for all $t \geq 1$ and $j \in [n]$. For any input $\boldsymbol{x} \in \{0, 1\}^n$, this sequence converges in finite time to a fixed point Bruck (1990). We define the input-output map of the Hopfield network, denoted $H : \{0, 1\}^n \times \Theta \to \{0, 1\}^n$ as follows: given parameters $\boldsymbol{\theta}$ and an input $\boldsymbol{x}$, the output $H_\theta(\boldsymbol{x}; \boldsymbol{\theta})$ is the attractor or fixed point of 2 reached when initialized with $\boldsymbol{x}(0) = \boldsymbol{x}$. If $H(\boldsymbol{x}; \boldsymbol{\theta}) = \boldsymbol{x}$, then $\boldsymbol{x}$ is a fixed point of the recurrence dynamics, furthermore in this setting we say that the function $H_\theta(\cdot) := H(\cdot\, ; \boldsymbol{\theta})$ has *memorized* $\boldsymbol{x}$. A sufficient condition for $H_\theta$ to memorize $\boldsymbol{x}$ is $E(\boldsymbol{x}; \boldsymbol{\theta}) < E(\boldsymbol{x}'; \boldsymbol{\theta})$ for all $\boldsymbol{x}' \in \mathcal{N}(\boldsymbol{x})$, where here $\mathcal{N}(\boldsymbol{x})$ denotes the set of all binary vectors a Hamming distance of exactly one from $\boldsymbol{x}$. Under this condition we say that $H_\theta$ *strictly memorizes* $\boldsymbol{x}$. We also denote the action of a permutation matrix $\boldsymbol{P} \in \mathcal{P}_n$ on the parameters of a Hopfield network as $\boldsymbol{P\theta} := (\boldsymbol{P}^T \boldsymbol{W} \boldsymbol{P}, \boldsymbol{P}^T \boldsymbol{b})$.

**Training and memorization:** let $\mathcal{S} \subset \{0,1\}^n$, we say that $H_\theta$ memorizes $\mathcal{S}$ if it memorizes all $\boldsymbol{x} \in \mathcal{S}$. There are many methods Hertz et al. (1991); Tolmachev & Manton (2020) that have been proposed to train networks to memorize a set $\mathcal{S}$ (Appendix A.2). We focus on minimization of the energy flow (MEF) Hillar et al. (2012); Hillar & Tran (2018); Hillar et al. (2021) and study its implicit bias. If $\boldsymbol{x} \in \{0,1\}^n$ and $j \in [n]$, let $\boldsymbol{x}^{(j)} \in \{0,1\}^n$ satisfy $x_l \neq x_l^{(j)}$ iff $l = j$. We define the *energy flow* loss as:

$$L(\boldsymbol{\theta}; \mathcal{S}) = \sum_{\boldsymbol{x} \in \mathcal{S}} \sum_{j=1}^{n} \exp\left(E(\boldsymbol{x}; \boldsymbol{\theta}) - E(\boldsymbol{x}^{(j)}; \boldsymbol{\theta})\right). \tag{3}$$

For any given set of points $\mathcal{S}$, note that $L$ is nonnegative, infinitely differentiable, and is convex in $\Theta$. As a result, minimizing $L$ is a convex problem to which a wide variety of numerical techniques can be applied, including, but not limited to, variants of gradient descent (GD) as well as (approximate) second order methods, such as L-BFGS Liu & Nocedal (1989). As long as $\mathcal{S}$ can be memorized, then sufficiently minimizing 3 will result in a network which memorizes $\mathcal{S}$. For further details on MEF learning for discrete recurrent neural networks, we refer the reader to Hillar et al. (2021) and its inspiration from the theory of density estimation Sohl-Dickstein et al. (2011).

## 3 Implicit bias, minimum norm memorizers and generalization

In this section, and prior to specializing to study datasets drawn from isomorphism classes of graphs, we connect memorization to solving a linear program and identify the implicit bias of MEF. This enables us to provide generalization guarantees for memorization in Hopfield networks as per Theorem 3.2. Given $\boldsymbol{\theta} = (\boldsymbol{W}, \boldsymbol{b}) \in \Theta$, for any $j \in [n]$ we define the vector $\boldsymbol{\theta}_j = [\boldsymbol{w}_j, b_j] \in \mathbb{R}^{n+1}$. Overloading our notation, we also use $\boldsymbol{\theta} = [\boldsymbol{\theta}_1, \boldsymbol{\theta}_2 ... \boldsymbol{\theta}_n] \in \mathbb{R}^{n,n+1}$ to refer to the flattened vector of all the network parameters $(\boldsymbol{W}, \boldsymbol{b})$. For any $\boldsymbol{x} \in \{0,1\}^n$ let $\boldsymbol{z}(\boldsymbol{x}) = [\boldsymbol{x}, 1] \in \{0,1\}^{n+1}$ and $y_j(\boldsymbol{x}) = 1 - 2x_j \in \{\pm 1\}$. Using this notation it is well known that the energy difference between a point and one of its neighbors is

$$E(\boldsymbol{x}^{(j)}; \boldsymbol{\theta}) - E(\boldsymbol{x}; \boldsymbol{\theta}) = y_j(\boldsymbol{x})\langle \boldsymbol{z}(\boldsymbol{x}), \boldsymbol{\theta}_j \rangle, \tag{4}$$

see Appendix A.1 for further details. As a consequence, parameters which strictly memorize a set $\mathcal{S} \subseteq \{0,1\}^n$ must satisfy a system of linear inequalities: in particular, there must exist some $\epsilon > 0$, referred to as the functional margin, such that $y_j(\boldsymbol{x})\langle \boldsymbol{z}(\boldsymbol{x}), \boldsymbol{\theta}_j \rangle \geq \epsilon$ for all $\boldsymbol{x} \in \mathcal{S}$ and $j \in [n]$. Clearly the energy function (1) is quadratic in the inputs $\boldsymbol{x}$ but linear in the parameters, $E(\boldsymbol{x}; a\boldsymbol{\theta}) = aE(\boldsymbol{x}, \boldsymbol{\theta})$. Moreover, this implies the energy is positively homogeneous of degree 1 in the parameters and as a result the set of attractors of a Hopfield network is invariant under positive rescaling of the parameters. Without loss of generality, we therefore select a functional margin of one and define the feasible set of parameters up to positive rescaling as

$$\mathcal{F}_\theta(\mathcal{S}) = \{\boldsymbol{\theta} \in \Theta : y_j(\boldsymbol{x})\langle \boldsymbol{z}(\boldsymbol{x}), \boldsymbol{\theta}_j \rangle \geq 1 \ \ \forall \boldsymbol{x} \in \mathcal{S}, \forall j \in [n]\} \tag{5}$$

In addition, the inequality constraints that define $\mathcal{F}(\mathcal{S})$ can be written with respect to a single vector of unique parameters, which we denote $\boldsymbol{\omega}$. To this end let $p = \frac{n(n-1)}{2}$ and $q = \frac{n(n+1)}{2}$. There exists a $\boldsymbol{V} \in \{0, 1, 1/\sqrt{2}\}^{n(n+1) \times q}$ such that for any $\boldsymbol{\theta} \in \Theta$ there exists an $\boldsymbol{a} \in \mathbb{R}^p$ with $\boldsymbol{\omega} = [\sqrt{2}\boldsymbol{a}, \boldsymbol{b}] \in \mathbb{R}^q$, such that $\boldsymbol{\theta} = \boldsymbol{V}\boldsymbol{\omega}$. In short, $\boldsymbol{V}$ copies the unique elements, i.e., the upper triangular elements of $\boldsymbol{W}$ and the biases $\boldsymbol{b}$, into their appropriate locations in the flattened vector $\boldsymbol{\theta}$. For any $j \in [n]$ let $\boldsymbol{V}_j \in \mathbb{R}^{(n+1) \times q}$ denote the submatrix of rows of $\boldsymbol{V}$ such that $\boldsymbol{\theta}_j = \boldsymbol{V}_j \boldsymbol{\omega}$. For any $\boldsymbol{x} \in \{0,1\}^n$ and $j \in [n]$ let $\boldsymbol{u}_j(\boldsymbol{x}) = y_j(\boldsymbol{x})\boldsymbol{V}_j^T \boldsymbol{z}(\boldsymbol{x})$. Then each constraint can be re-written as $y_j(\boldsymbol{x})\langle \boldsymbol{z}(\boldsymbol{x}), \boldsymbol{\theta}_j \rangle = \langle \boldsymbol{u}_j(\boldsymbol{x}), \boldsymbol{\omega} \rangle$ and thus we can equivalently define the feasible set as

$$\mathcal{F}_\omega(\mathcal{S}) = \{\boldsymbol{\omega} \in \mathbb{R}^q : \langle \boldsymbol{u}_j(\boldsymbol{x}), \boldsymbol{\omega} \rangle \geq 1 \ \ \forall \boldsymbol{x} \in \mathcal{S}, \forall j \in [n]\} \tag{6}$$

Inspecting (6), clearly strict memorization of a dataset is equivalent to solving a linear program (LP) and therefore any algorithm which successfully memorizes $\mathcal{S}$ is implicitly solving an LP. Moreover, these algorithms may have an *implicit bias* towards feasible points or solutions which satisfy other conditions or criteria. A popular and well studied example is the feasible point with the smallest norm: identifying this requires solving a quadratic program (QP) or, more specifically, a hard margin support vector machine (HSVM) problem. In

particular, note that if $\boldsymbol{\theta} = \boldsymbol{V}\boldsymbol{\omega}$ where $\boldsymbol{\omega} = [\sqrt{2}\boldsymbol{a}, \boldsymbol{b}]$ then $\|\boldsymbol{\theta}\|^2 = \|\boldsymbol{W}\|_F^2 + \|\boldsymbol{b}\|^2 = \|\sqrt{2}\boldsymbol{a}\|^2 + \|\boldsymbol{b}\|^2 = \|\boldsymbol{\omega}\|^2$. As a result, finding the minimum norm feasible point for a set $\mathcal{S} \subset \{0, 1\}^n$ is equivalent to solving

$$\text{HSVM}(\mathcal{S}) = \underset{\boldsymbol{\omega} \in \mathbb{R}^q}{\arg\min} \|\boldsymbol{\omega}\|^2 \quad s.t. \quad \boldsymbol{\omega} \in \mathcal{F}_\omega(\mathcal{S}). \tag{7}$$

The key takeaway of this section is that minimizing (3) with gradient descent (GD) is implicitly biased in direction towards the solution of (7), i.e., norm-minimization. To this end, first observe that (3) can be re-parameterized as

$$L(\boldsymbol{\theta}; \mathcal{S}) = \sum_{\boldsymbol{x} \in \mathcal{S}} \sum_{j=1}^n \exp\left( E(\boldsymbol{x}; \boldsymbol{\theta}) - E(\boldsymbol{x}^{(j)}; \boldsymbol{\theta}) \right) = \sum_{\boldsymbol{x} \in \mathcal{S}} \sum_{j=1}^n \exp\left( -\langle \boldsymbol{u}_j(\boldsymbol{x}), \boldsymbol{\omega} \rangle \right) =: L(\boldsymbol{\omega}; \mathcal{S}).$$

Consider now updates to the parameters of the Hopfield network using GD: in particular, given initial parameters $\boldsymbol{\omega}(0)$ and step-size $\eta > 0$, for all $t \geq 0$ let

$$\boldsymbol{\omega}(t+1) = \boldsymbol{\omega}(t) + \eta \sum_{\boldsymbol{x} \in \mathcal{S}} \sum_{j=1}^n \exp\left( -\boldsymbol{u}_j(\boldsymbol{x})^T \boldsymbol{\omega}(t) \right) \boldsymbol{u}_j(\boldsymbol{x}). \tag{8}$$

Using (Soudry et al., 2018, Thm.3), it can be proved that this sequence of GD iterates converges in direction to the solution of (7). Indeed, (Soudry et al., 2018, Thm.3) directly implies the following in our setting.

**Corollary 3.1.** *((Soudry et al., 2018, Thm.3))* *Assume $\mathcal{S}$ can be strictly memorized, let $\boldsymbol{\omega}^* = \text{HSVM}(\mathcal{S})$, $\boldsymbol{\omega}(0) \in \mathbb{R}^q$ be arbitrary and $\boldsymbol{\omega}(t)$ be generated for all $t \in \mathbb{N}_{\geq 1}$ as per (8). There exists a choice of step size $\eta$ such that $\boldsymbol{\omega}(t) = \boldsymbol{\omega}^* \log(t) + \rho(t)$ for all $t \in \mathbb{N}_{\geq 1}$, where $\rho(t)$ grows as $\|\rho(t)\| = O(\log(\log(t)))$. Moreover, $\lim_{t\to\infty} \frac{\boldsymbol{\omega}(t)}{\|\boldsymbol{\omega}(t)\|} = \frac{\boldsymbol{\omega}^*}{\|\boldsymbol{\omega}^*\|}$.*

Informally, Corollary 3.1 states that the solution returned by minimizing the energy flow with gradient descent (MEF-GD) after exponentially many iterations is a close approximation directionally to the solution returned by solving the HSVM problem (7). We now derive generalization bounds both for the HSVM solution and MEF with GD.

**Theorem 3.2.** *Let $D \subset \{0, 1\}^n$ be a set which can be strictly memorized and assume $\|\boldsymbol{x}\|_0 \leq m \in \mathbb{N}_{\geq 1}$ for all $\boldsymbol{x} \in D$. Let $\mathcal{D}$ be a probability distribution on $D$, and consider a random sample $\mathcal{S}_N = (\boldsymbol{x}_i)_{i \in [N]}$, where $\boldsymbol{x}_i \sim \mathcal{D}$ are mutually i.i.d. Let $\hat{\boldsymbol{\omega}} = \text{HSVM}(\mathcal{S}_N)$, $\boldsymbol{\omega}^* = \text{HSVM}(D)$, $\boldsymbol{\omega}(0) \in \mathbb{R}^q$ be arbitrary and $\boldsymbol{\omega}(t)$ be generated for all $t \in \mathbb{N}_{\geq 1}$ as per (8), $\delta, \epsilon \in (0, 1)$ and assume $\boldsymbol{x} \sim \mathcal{D}$ is sampled independent of $\mathcal{S}_N$. If $N \gtrsim \epsilon^{-2} n \|\boldsymbol{\omega}^*\|^2 m \log(1/\delta)$ then both*

$$\mathbb{P}(H(\boldsymbol{x}; \boldsymbol{V}\hat{\boldsymbol{\omega}}) \neq \boldsymbol{x}) \leq \epsilon \quad and \quad \mathbb{P}(H(\boldsymbol{x}; \boldsymbol{V}\boldsymbol{\omega}(t)) \neq \boldsymbol{x}) = \tilde{O}\left( \frac{\sqrt{m}\|\boldsymbol{\omega}^*\|}{\log(t)} \right) + \epsilon$$

*hold with probability at least $1 - \delta$ over the sample $\mathcal{S}_N$.*

Note, the $\tilde{O}(\cdot)$ notation hides a factor of $\log(\log(t))$ arising in (Soudry et al., 2018, Thm.5). To prove Theorem 3.2 we combine a vector contraction inequality (Maurer, 2016, Corollary 1) with Rademacher bounds, see e.g., (Mohri et al., 2018, Theorem 3.3), we refer the reader to Appendix B.1 for a full proof. It is worth emphasizing that Theorem 3.2 implies that any dataset $D$ which can be strictly memorized, can be at least *nearly* strictly memorized using only a polynomial number of samples. In Section 4.3 we take preliminary steps towards relaxing this statement, i.e., from memorizing samples drawn from $\mathcal{D}$ with high probability, to memorizing the set $D$ itself with high probability. Finally, we comment that the MEF bound implies gradient descent may require exponentially many iterations to converge directionally to the max-margin solution. In practice we emphasize that this is a tail phenomenon: once the weights are approximately aligned, all points are classified with a significant margin and their loss contributions become exponentially small.

## 4 Storing isomorphism classes of graphs

### 4.1 Graph encoding

Let $\mathcal{G}_v$ denote the set of all simple, undirected graphs on $v \in \mathbb{N}$ vertices. Recall two graphs $G = (V, E)$, $G' = (V', E')$ are isomorphic, which we denote $G \cong G'$, if there exists a bijection $\phi : V \to V'$ such that

$(\phi(\nu_1), \phi(\nu_2)) \in E'$ if and only if $(\nu_1, \nu_2) \in E$. We refer to such a $\phi$ as an isomorphism between $G$ and $G'$. Note when $V = V'$, as is the case here since $V = V' = [v]$, then $\phi$ is a permutation. The *isomorphism class* of a graph $G \in \mathcal{G}_v$ is defined as $\mathcal{I}(G) := \{G' \in \mathcal{G}_v : G' \cong G\}$. Let $\mathcal{V}_2$ denote the set of unordered pairs of $[v]$, $n := |\mathcal{V}_2| = \binom{v}{2}$ and $\text{Ind} : \mathcal{V}_2 \to [n]$ be a bijection which indexes the elements of $\mathcal{V}_2$.

**Definition 4.1** (Edge representation of a graph). Let $\mathcal{E}_{rep} : \mathcal{G}_v \to \{0,1\}^n$ be defined as follows: if $G = ([v], E) \in \mathcal{G}_v$ and $\boldsymbol{x} = \mathcal{E}_{rep}(G)$ then, for all $j \in [n]$, $x_j := \mathbb{1}(\text{Ind}^{-1}(j) \in E)$. We refer to $\boldsymbol{x}$ as the edge representation of $G$.

To be clear, $\mathcal{E}_{rep}$ is a bijection which assigns each graph to a binary vector of dimension $n$ whose support defines the vertex pairs present in the edge set of the graph in question. Any vertex permutation induces an edge permutation.

**Definition 4.2** (Edge permutation induced by a vertex permutation). Let $\phi : [v] \to [v]$ be a permutation. The edge permutation $\pi_\phi : [n] \to [n]$ induced by $\phi$ is defined as follows: if $\text{Ind}^{-1}(j) = (\nu_1, \nu_2)$ then $\pi_\phi(j) = \text{Ind}((\phi(\nu_1), \phi(\nu_2)))$. We denote the subset of these edge permutations as $\Pi_n^\Phi$ and the corresponding subset of permutation matrices as $\Phi_n$.

We now claim the following: first $\Phi_n$ is a subgroup of $\mathcal{P}_n$, second if two graphs $G, G' \in \mathcal{G}_v$ are isomorphic then there is a vertex induced edge permutation which maps between their edge representations, and third, for any $G \in \mathcal{G}_v$ we have $\mathcal{E}_{rep}(\mathcal{I}(G)) = \text{Orb}(\mathcal{E}_{rep}(G), \Phi_n)$. For proofs of these claims we refer the reader to Appendix A.3. Two edges are said to be *adjacent* if they share a vertex in common: more specifically, if $j, l \in [n]$ then $j$ and $l$ are *adjacent*, which we denote $j \sim l$, if and only if $|\text{Ind}^{-1}(j) \cap \text{Ind}^{-1}(l)| = 1$, otherwise we say $j$ and $l$ are not adjacent, which we denote $j \nsim l$. We now identify a subset of edge permutations which are characterized by preserving edge adjacency.

**Definition 4.3** (Edge adjacency preserving permutation). A permutation $\pi : [n] \to [n]$ *preserves edge adjacency* if $\pi(j) \sim \pi(l)$ if and only if $j \sim l$. We denote such permutations as $\Pi_n^\mathcal{Q}$ and the corresponding set of permutation matrices as $\mathcal{Q}_n$.

Similar to $\Phi_n$, this subset forms a subgroup of $\mathcal{P}_n$. Moreover $\Phi_n$ is a subgroup of $\mathcal{Q}_n$ and as a result $\mathcal{E}_{rep}(\mathcal{I}(G)) \subset \text{Orb}(\mathcal{E}_{rep}(G), \mathcal{Q}_n)$. Again we refer the reader to Appendix A.3 for further details. As a result, the edge representations of the isomorphism class of a graph are a subset of the orbit of the edge representation of the graph in question under edge adjacency preserving permutations.

## 4.2 Experiments on graph data

To experimentally assess storage across isomorphism classes we study several classes of graphs; namely clique, bipartite, Paley and Johnson graphs. These families are standard examples in graph theory and algebraic graph theory, see for example Godsil & Royle (2001). Clique graphs, or more specifically $k$-cliques, contain a fully connected subset of $k$ vertices while the remaining $v - k$ vertices are isolated. Bipartite graphs split the $v$ vertices into two equally sized groups with all possible inter-group edges present and no intra-group edges. Paley graphs connect vertices $l$ and $j$ when $(l - j)$ is a quadratic residue mod $v$, as per NetworkX Developers; cf. Bollobás (2001). For integers $n, r$, the Johnson graph $J(n, r)$ has as vertices the $r$-element subsets of $[n]$, with two vertices adjacent when the corresponding subsets differ in exactly one element [2]. These graphs are natural test cases because they are regular, highly symmetric and distance-regular while still exhibiting a nontrivial combinatorial structure that is distinct from cliques, bipartite graphs and Paley graphs. We remark that extensive experiments for cliques are already provided in Hillar & Tran (2018); Hillar et al. (2021); we include them here again for comparison and completeness. We also remark that we selected these classes due to the ease with which we are able to sample from and enumerate them, however, we emphasize that these families are representative rather than special. Indeed, we observe similar behavior for many other graph isomorphism classes, including random graphs. We refer the reader to our reproducibility statement which includes a link to our code. In addition, further experiments and preliminary results on topics including the Hidden Clique Problem (Fig. 6), comparison versus Dense Associative Memories (DAMs) (Fig. 7), orbit generalization for other graph families (Figs 8, 9), and double descent phenomena (Fig. 10) can be found in Appendix C.

---

[2] Equivalently, when their intersection has size $(r - 1)$.

Fig. 1 shows test accuracy versus training sample size, with mean and min–max over 10 trials, for Hopfield networks trained by MEF, Perceptron, and Delta (the latter two used only as baselines; see Appendix A.2). For small graphs ($v = 8$) we enumerate the full isomorphism class and report the true accuracy, i.e., the fraction of the class memorized. For larger graphs ($v = 20$), accuracy is estimated on an independent random sample of 1000 graphs. We highlight two observations: (i) MEF appears to reach higher test accuracy with fewer samples relative to the other methods, despite all methods perfectly memorizing the training set. This suggests differing implicit biases or implicit bias strengths. (ii) For MEF and Delta, the sample size needed to memorize an isomorphism class is tiny relative to the class size, aligning with the findings in Hillar & Tran (2018); Hillar et al. (2021). Furthermore, the number of iterations was capped at 1000, suggesting that the exponential dependency in Theorem 3.2 is highly pessimistic. We note that analogous experiments for other graph types are provided in Appendix C.

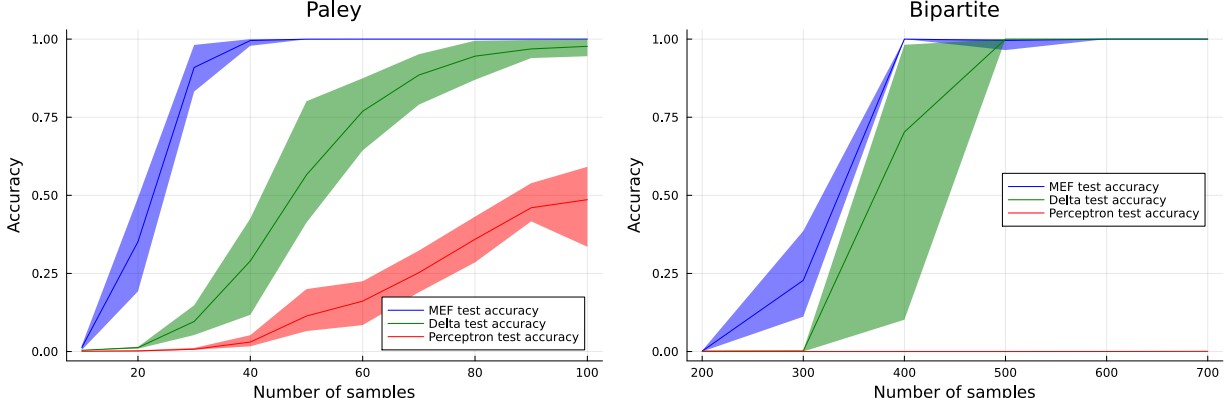

Figure 1: **Test accuracy vs. training sample size for isomorphism classes at two scales.** Top row: $v = 8$, isomorphism class size for Paley is 2520. Bottom row: $v = 20$, isomorphism class size for bipartite is 92,378. Curves show mean and min-max over 10 trials. Networks are trained with Perceptron, Delta (MSE), and MEF learning rules.

Fig. 2 estimates and compares the specific polynomial sample complexity of learning $k$-cliques versus Paley graphs. We do this in order to highlight that different isomorphism classes may be harder or easier to learn depending on their connectivity structure. For each graph size $v$ we record $s_{50}$, which we define as the smallest training sample size for which MEF attains $\geq 50\%$ average test accuracy on test samples of size 1000 averaged over 10 trials. The left subplot shows $s_{50}$ vs. $v$. The right subplot shows a log-log fit. Assuming $s_{50} = Cv^p$ for some constant $C \in \mathbb{R}_{>0}$, this allows us to estimate $p$ via linear regression. While Paley graphs need $N = \tilde{\Omega}(v^{2+\epsilon})$, cliques require $N = \tilde{\Omega}(v^{1+\epsilon})$ (note here we use $\epsilon \in (0, 1)$ to denote a small error term). Thus, although Hopfield networks can memorize all classes (Lemma 4.7), the specific sample complexity appears to vary with graph connectivity.

Fig. 3 shows histograms of the elements of weight matrices of Hopfield networks trained on isomorphism classes of graphs, namely Bipartite and Johnson graphs, for three different sample sizes. Indeed, it is apparent that as the sample size grows the parameters returned by the optimizer converge onto a distinct subspace: we identify this subspace as the parameters invariant to the underlying group action of the data in Lemma 4.6 below. Heatmaps further supporting this conclusion can be observed in Figure 5 in Appendix C.

### 4.3 Invariant parameters

In what follows let $\Gamma_n$ denote an arbitrary subgroup of $\mathcal{P}_n$. For any $\boldsymbol{Q} \in \mathcal{P}_n$ and $\boldsymbol{\theta} = (\boldsymbol{W}, \boldsymbol{b}) \in \boldsymbol{\Theta}$ recall that we define the action of $\boldsymbol{Q}$ on the parameter $\boldsymbol{\theta}$ as $\boldsymbol{Q\theta} := (\boldsymbol{Q}^T \boldsymbol{W} \boldsymbol{Q}, \boldsymbol{Q}^T \boldsymbol{b})$. Let $\boldsymbol{Q} \in \mathcal{P}_n$, $\boldsymbol{\theta} = (\boldsymbol{W}, \boldsymbol{b}) \in \boldsymbol{\Theta}$ and $\boldsymbol{\theta}' = \boldsymbol{Q\theta} = (\boldsymbol{W}', \boldsymbol{b}')$. Note $\boldsymbol{W}'^T = (\boldsymbol{Q}^T \boldsymbol{W} \boldsymbol{Q})^T = \boldsymbol{Q}^T \boldsymbol{W} \boldsymbol{Q} = \boldsymbol{W}'$ and for all $j \in [n]$ we have $W'_{jj} = \boldsymbol{e}^T_{\pi(j)} \boldsymbol{W} \boldsymbol{e}_{\pi(j)} = W_{\pi(j)\pi(j)} = 0$. As a result $\boldsymbol{W}' \in \mathrm{Sym}^n_0$, in addition trivially $\boldsymbol{b}' \in \mathbb{R}^n$ and therefore $\boldsymbol{\theta}' \in \boldsymbol{\Theta}$. As a result, the action of $\mathcal{P}_n$, or any subgroup $\Gamma_n$ of $\mathcal{P}_n$, on $\boldsymbol{\Theta}$ is well defined.

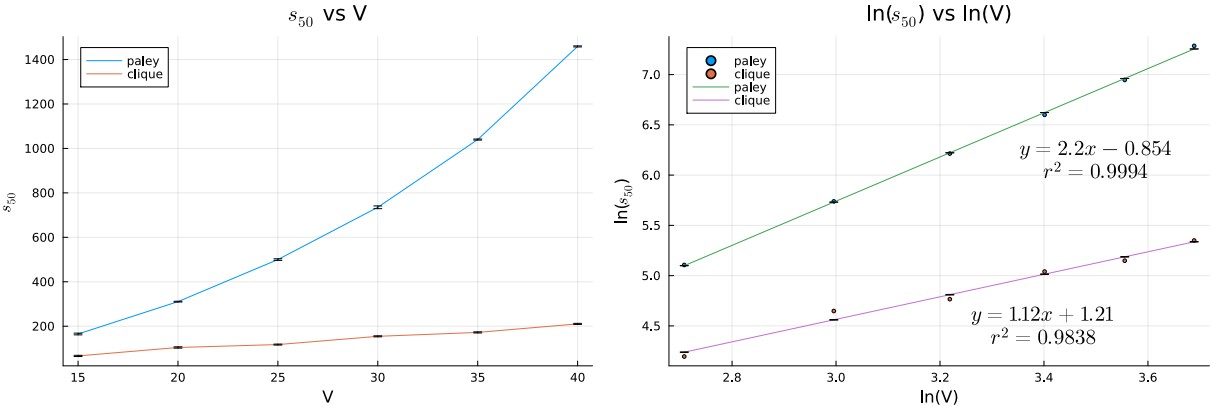

Figure 2: **Sample complexity scaling.** Plots showing the number of samples $s_{50}$ required for a Hopfield network trained via MEF using accelerated gradient descent to memorize 50% of a random sample of 1000 graphs drawn from clique and Paley graph isomorphism classes on $v$ vertices. On the right we plot $\ln(s_{50})$ vs $\ln(v)$ and compute the lines of best fit.

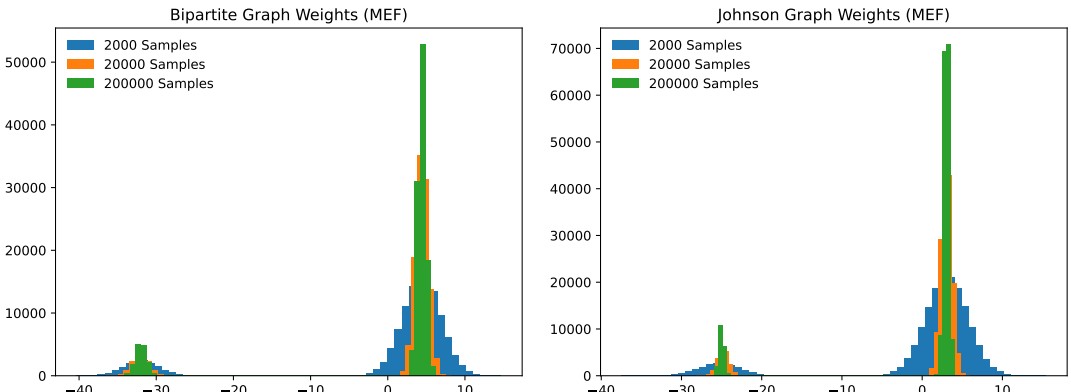

Figure 3: **Learned weights approach invariant subspace as sample size increases.** Histograms of MEF-learned weights over 3 sample counts. Top row: Bipartite graph with $v = 32$, $k = 16$ (496-bit). Bottom row: Johnson graph $J(7,3)$ on $v = 35$ vertices (595-bit). Parameters are normalized so that thresholds (not shown) have mean absolute value 1.

**Definition 4.4** (Parameter invariance to the action of a subgroup). A parameter $\boldsymbol{\theta} \in \Theta$ is invariant with respect to $\Gamma_n$ iff for all $\boldsymbol{Q} \in \Gamma_n$ then $\boldsymbol{Q}\boldsymbol{\theta} = \boldsymbol{\theta}$. We denote the set of these parameters as $\Psi(\Gamma_n)$.

Recall $E(\boldsymbol{Q}\boldsymbol{x}; \boldsymbol{\theta}) = \frac{1}{2}\boldsymbol{x}^T(\boldsymbol{Q}^T\boldsymbol{W}\boldsymbol{Q})\boldsymbol{x} + (\boldsymbol{Q}^T\boldsymbol{b})^T\boldsymbol{x}$. If $\boldsymbol{\theta} \in \Psi(\Gamma_n)$, then for any $\boldsymbol{x} \in \{0,1\}^n$ and $\boldsymbol{Q} \in \Gamma_n$ we have

$$E(\boldsymbol{Q}\boldsymbol{x}; \boldsymbol{\theta}) = E(\boldsymbol{x}; \boldsymbol{Q}\boldsymbol{\theta}) = E(\boldsymbol{x}; \boldsymbol{\theta}). \tag{9}$$

We refer to (9) as the intertwining property of the energy function. Using this property, the following lemma extends energy difference bounds between neighbors from a point to an orbit.

**Lemma 4.5.** *Let $\boldsymbol{x}_0 \in \{0,1\}^n$ and $\boldsymbol{\theta} \in \Psi(\Gamma_n)$. For $\delta \in \mathbb{R}$, if $E(\boldsymbol{x}_0^{(j)}; \boldsymbol{\theta}) - E(\boldsymbol{x}_0; \boldsymbol{\theta}) \geq 1 - \delta$ for all $j \in [n]$, then for all $\boldsymbol{x} \in \mathrm{Orb}(\boldsymbol{x}_0, \Gamma_n)$ it follows that $E(\boldsymbol{x}^{(j)}; \boldsymbol{\theta}) - E(\boldsymbol{x}; \boldsymbol{\theta}) \geq 1 - \delta$ for all $j \in [n]$.*

For a proof of this lemma, as well as the other results presented in this section, we refer the reader to Appendix B.2. A key implication of Lemma 4.5 is that if $\boldsymbol{\theta} \in \Psi(\Gamma_n)$ strictly memorizes $\boldsymbol{x}_0 \in \{0,1\}^n$, then $\boldsymbol{\theta}$ also strictly memorizes $\mathrm{Orb}(\boldsymbol{x}_0, \Gamma_n)$. We now show that invariance with respect to edge adjacency preserving permutations, of which graph isomorphisms are a subset, corresponds to a particular rank three subspace of the parameters.

**Lemma 4.6.** *Let $F : \mathbb{R}^3 \to \Theta$ denote the linear map defined as follows: if $(\boldsymbol{W}, \boldsymbol{b}) = F(\boldsymbol{\beta})$ then for all $i, j \in [n]$ we have $w_{ij} = 0$ if $i = j$, $w_{ij} = \beta_1$ if $i \sim j$, $w_{ij} = \beta_2$ if $i \nsim j$ and $b_j = \beta_3$. Then $\Psi(\mathcal{Q}_n) = F(\mathbb{R}^3)$ where $F(\mathbb{R}^3)$ denotes the image of $F$.*

By inspection, the parameter patterns observed in Figs. 3, 5, 6b, 9bd for MEF appear to approximately lie on the invariant subspace identified in Lemma 4.6. This suggests, given a sufficiently large training sample, that there is an implicit bias not just towards small norm solutions, but also those that are at least approximately invariant. Following this observation, a natural question to ask is whether or not parameters lying on this subspace can memorize any graph.

**Lemma 4.7.** *For $m \in [0, n]$, let $\boldsymbol{\beta} = [2, 2, 1 - 2m] \in \mathbb{R}^3$ and $\boldsymbol{\theta} = F(\boldsymbol{\beta})$. Then $E(\boldsymbol{x}^{(j)}; \boldsymbol{\theta}) - E(\boldsymbol{x}; \boldsymbol{\theta}) \geq 1$ for all $j \in [n]$ and for all $\boldsymbol{x} \in \{0, 1\}^n$ satisfying $\|\boldsymbol{x}\|_0 = m$.*

Combining Lemmas 4.7 and 4.5 we conclude that any graph isomorphism class can be strictly memorized by a Hopfield network. We also note that the only statistic used by the construction in Lemma 4.7 is the sparsity of the representation: in fact, this construction memorizes $\boldsymbol{x} \in \{0, 1\}^n$ iff $\|\boldsymbol{x}\|_0 = m$. As a result, if our goal is to memorize an isomorphism class while avoiding spurious memories this is a poor parameter candidate. In addition, assuming $m = \Theta(n)$, the norm of this construction grows as $\Theta(v^2)$. For specific graph isomorphism classes solutions with a far smaller norm exist. As an example we consider $k$-cliques: for typographical ease we denote the set of binary representations of $k$-cliques on $v$ vertices as $\mathcal{C}_{v,k}$.

**Lemma 4.8.** *If $\boldsymbol{\beta} = [-10/k, 38/k^2, 0] \in \mathbb{R}^3$, $\boldsymbol{\theta} = F(\boldsymbol{\beta})$, and $k \geq 5$, then the following hold.*

1. *$E(\boldsymbol{x}^{(j)}; \boldsymbol{\theta}) - E(\boldsymbol{x}; \boldsymbol{\theta}) \geq 1$ for all $\boldsymbol{x} \in \mathcal{C}_{v,k}$ and all $j \in [n]$.*

2. *If $k = cv$ for some constant $c \in (0, 1]$, then there exists a constant $C > 0$ such that $\|\boldsymbol{\theta}\|^2 \leq Cv$.*

Lemma 4.8 shows that a parameter exists which strictly memorizes $\mathcal{C}_{v,k}$ with norm only $O(\sqrt{v})$ rather than $\Theta(v^2)$. The construction in Lemma 4.8 also does not memorize all $m$-sparse binary vectors. For example, fixing some $j \in [n]$, suppose $\boldsymbol{x}$ is such that $\|\boldsymbol{x}\|_0 = m \leq 2(v - 2)$ and for all $l \in \text{supp}(\boldsymbol{x})$ we have $l \sim j$. Then $\boldsymbol{u}_j(\boldsymbol{x})^T \boldsymbol{\omega} = -10m/k$ and as a result $\boldsymbol{x}$ is not strictly memorized. We speculate that perhaps a correlation between the size of the norm and the number of spurious memories exists, but we leave a proper investigation to future work.

Before proceeding we pause to reflect on the implications of our results with respect to (Hillar et al., 2021, Conjecture 1). Under a linear density regime $k = cv$ where $c \in (0, 1)$ is a constant, then together Theorem 3.2, Lemma 4.7 and Lemma 4.8 imply that $k$-cliques on $v$ vertices can be strictly memorized with high probability as long as $N = \tilde{\Omega}(v^5)$. For simplicity assuming $cv$ is an integer, then using Stirling's approximation the critical ratio satisfies $\tilde{O}(v^5)/\binom{v}{cv} = \tilde{O}\left(2^{-vH(c)}v^{5.5}\right)$, where here at the risk of confusion we use $H$ to denote the binary entropy function. Clearly the critical ratio decays to zero at a rate which as $v \to \infty$ is dominated by the exponential term.

Our experiments and results thus far suggest that memorization of a graph isomorphism class occurs when the training sample is sufficiently large that the optimizer is forced to return a solution lying close to the invariant set $\Psi(\Phi_n) \subset F(\mathbb{R}^3)$. The following lemma establishes that the HSVM solution on the full isomorphism class, which as $N \to \infty$ is equivalent to the training sample with probability one, must be graph isomorphism invariant.

**Lemma 4.9.** *Let $\boldsymbol{x}_0 \in \{0, 1\}^n$ and $\Gamma_n$ denote a subgroup of $\mathcal{P}_n$ and assume $\text{Orb}(\boldsymbol{x}_0, \Gamma_n)$ can be strictly memorized. If $\boldsymbol{\theta}^* = \boldsymbol{V}\boldsymbol{\omega}^*$ where $\boldsymbol{\omega}^* = \text{HSVM}_\Theta(\text{Orb}(\boldsymbol{x}_0, \Gamma_n))$ then $\boldsymbol{\theta}^* \in \Psi(\Gamma_n)$.*

Following Lemma 4.9, we ask how many samples do we require in order to achieve at least approximate invariance? Deriving a sample complexity result is challenging, primarily due to the fact that the feasible set of the HSVM problem changes non-smoothly with respect to the training sample. Instead, and in order to gain intuition, we conclude this section by analyzing a related but simpler problem, which we refer to as the

average hard-margin support vector machine (AHSVM) problem. To this end, we define the following,

$$\mathcal{F}_A(\mathcal{S}) = \{\boldsymbol{\omega} \in \mathbb{R}^q : \frac{1}{|\mathcal{S}|} \sum_{\boldsymbol{x} \in \mathcal{S}} \langle \bar{\boldsymbol{u}}(\boldsymbol{x}), \boldsymbol{\omega} \rangle \geq 1\},$$

$$\text{AHSVM}(\mathcal{S}) = \underset{\boldsymbol{\omega} \in \mathbb{R}^q}{\arg \min} \frac{1}{2} \|\boldsymbol{\omega}\|^2 \ s.t. \ \boldsymbol{\omega} \in \mathcal{F}_A(\mathcal{S}).$$

The following lemma bounds the difference between the sample AHSVM solution and the population AHSVM solution in the context of a uniform distribution over an arbitrary $\mathcal{O} \subseteq \{0, 1\}^n$.

**Lemma 4.10.** *Let $\mathcal{O} \subseteq \{0, 1\}^n$ satisfy $\|\boldsymbol{x}\|_0 \leq m \in \mathbb{N}_{\geq 2}$ for all $\boldsymbol{x} \in \mathcal{O}$ and assume $\boldsymbol{\omega}^* = \text{HSVM}(\mathcal{O})$ is feasible. Consider a random sample $\mathcal{S} = (\boldsymbol{x}_i)_{i=1}^N$ where $\boldsymbol{x}_i \sim U(\mathcal{O})$ are mutually i.i.d. and define $\boldsymbol{\omega}_{\mathcal{O}} = \text{AHSVM}(\mathcal{O})$ and $\boldsymbol{\omega}_{\mathcal{S}} = \text{AHSVM}(\mathcal{S})$. For $\delta \in (0, 1]$ and $\epsilon \in \mathbb{R}_{>0}$, if $N \gtrsim \epsilon^{-2} \|\boldsymbol{\omega}^*\|^4 m \log(1/\delta)$ then $\|\boldsymbol{\omega}_{\mathcal{S}} - \boldsymbol{\omega}_{\mathcal{O}}\| \leq \epsilon$ with probability at least $1 - \delta$.*

Now let $\text{Proj}_{\Psi(\Phi_n)}^{\perp}(\boldsymbol{\theta})$ denote the projection onto the subspace orthogonal to $\Psi(\Phi_n)$. Together, Lemmas 4.8, 4.10 and B.6 characterize proximity of the AHSVM solution for a $k$-clique sample to the invariant subspace.

**Corollary 4.11.** *Assume $k = cv \geq 5$ for some constant $c \in (0, 1)$ and let $\mathcal{S}_N = (\boldsymbol{x}_i)_{i \in [N]}$, where $\boldsymbol{x}_i \sim U(\mathcal{C}_{v,k})$ are mutually i.i.d. Let $\boldsymbol{\omega} = \text{AHSVM}(\mathcal{S}_N)$ and $\boldsymbol{\theta} = \boldsymbol{V}\boldsymbol{\omega}$. For $\delta \in (0, 1)$, if $N \gtrsim \epsilon^{-2} v^4 \log(1/\delta))$ then $\|\text{Proj}_{\Psi(\Phi_n)}^{\perp}(\boldsymbol{\omega})\| \leq \epsilon$ with probability at least $1 - \delta$.*

Corollary 4.11 illustrates that, at least for the AHSVM problem, we can get arbitrarily close to the invariant subspace with high probability using a sample size cubic in the number of vertices. We emphasize that even in the AHSVM setting bounding the distance from the learned parameters to the invariant subspace is challenging. We leave further refinement of these results as well as a derivation of an analogous one for the HSVM problem to future work.

# 5 Conclusions, Limitations and Future Work

In summary, this work shows that classical Hopfield networks can exploit symmetry in graph-structured data. By reformulating strict memorization as a linear feasibility problem, we identify norm efficiency as a mechanism linking MEF training, max-margin solutions, and sample-efficient orbit generalization. For graph isomorphism classes, the invariant parameter subspace provides a compact explanation for the observed few-shot-to-orbit phenomenon: as training samples increasingly constrain the feasible set, low-norm solutions are driven toward approximately invariant representations. These results suggest that even in minimal associative-memory models, implicit bias can substitute in part for architectural invariance.

This work has limits: while the full-orbit HSVM solution is invariant and, since the orbit is finite, an i.i.d. sample eventually contains the entire orbit almost surely, we do not yet provide a quantitative finite-sample convergence rate showing how quickly the HSVM or MEF solutions approach the invariant subspace. We also do not yet explain why some isomorphism classes appear to be easier to learn than others. Future work should quantify spurious fixed points and basin robustness, treat other subgroups and unions of orbits, handle noisy or non-uniform group data, extend the analysis to hypergraphs, and involve continuous and modern HNs. Towards achieving these goals, we highlight preliminary experimental findings detailed in Appendix C concerning invariant subspace convergence (Fig. 5), the Hidden Clique Problem (Fig. 6), comparison to DAMs (Fig. 7), generalization in other graph families (Figs. 8, 9), double descent phenomena (Fig. 10), and isomorphic graph checking (Algorithm 1).

**Reproducibility Statement:** Code able to reproduce our experimental results can be found in the following anonymous repository, `https://github.com/hopnetorbit/HopfieldNetworksIsomorphism`.

**Ethics Statement:** This work uses only synthetic, non-sensitive data, involves no human or animal subjects, carries minimal dual-use or environmental risks (modest compute).

**Broader Impact Statement:** This work contributes to the mathematical and conceptual foundations of machine learning and intelligent systems. Although the results are primarily theoretical, a better understanding of the underlying principles governing learning, generalization, and representation can inform the long-term development of more reliable, interpretable, and robust algorithms.

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

# A   Background

## A.1   Energy gap for binary vectors a hamming distance one apart

As discussed in Section 2, memorization is equivalent of a point is equivalent to ensuring an energy gap between it and its neighbors. Recall we define $\boldsymbol{x}^{(j)} \in \{0,1\}^n$ as the vector that differs from $\boldsymbol{x} \in \{0,1\}^n$ only at the $j$th location, and $\boldsymbol{z}(\boldsymbol{x}) = [\boldsymbol{x}, 1] \in \mathbb{R}^{n+1}$.

**Lemma A.1.** $E(\boldsymbol{x}^{(j)}; \boldsymbol{\theta}) - E(\boldsymbol{x}; \boldsymbol{\theta}) = y_j(\boldsymbol{x})\langle \boldsymbol{z}(\boldsymbol{x}), \boldsymbol{\theta}_j \rangle.$

*Proof.* By definition $x_l^{(j)} \neq x_l$ iff $l = j$. In addition, $x_j^{(j)} - x_j = 1 - 2x_j$ and, if $r \neq l$, then $x_l x_r \neq x_l^{(j)} x_r^{(j)}$ iff either $l = j$ and $r \neq j$, or $l \neq j$ and $r = j$. Furthermore, recall $\boldsymbol{W}$ is symmetric and $W_{jj} = 0$ for all $j \in [n]$. As a result,

$$
\begin{aligned}
E(\boldsymbol{x}^{(j)}; \boldsymbol{\theta}) - E(\boldsymbol{x}; \boldsymbol{\theta}) &= \frac{1}{2} \sum_{l,r \in [n]} W_{rl}(x_r^{(j)} x_l^{(j)} - x_r x_l) + \sum_{l=1}^{n} b_l(x_l^{(j)} - x_l) \\
&= \frac{1}{2} \sum_{r \in [n], r \neq j} W_{rj}(x_r^{(j)} x_j^{(j)} - x_r x_j) + \frac{1}{2} \sum_{l \in [n], l \neq j} W_{jl}(x_j^{(j)} x_l^{(j)} - x_j x_l) + b_j(1 - 2x_j) \\
&= \sum_{l \in [n], l \neq j} W_{jl}(x_j^{(j)} x_l^{(j)} - x_j x_l) + b_j(1 - 2x_j) \\
&= \sum_{l \in [n], l \neq j} W_{jl}(x_j^{(j)} - x_j)x_l + b_j(1 - 2x_j) \\
&= (1 - 2x_j)\left( \sum_{l \in [n]} W_{jl}x_l + b_j \right) \\
&= y_j(\boldsymbol{x})\langle \boldsymbol{z}(\boldsymbol{x}), \boldsymbol{\theta}_j \rangle.
\end{aligned}
$$

as claimed. □

## A.2   Learning algorithms for Hopfield Networks

We briefly describe several classical learning rules Hertz et al. (1991) that can be applied to find parameters in Hopfield networks. These methods typically trade off between biological plausibility and performance. We remark that this list is far from exhaustive; see Tolmachev & Manton (2020) for a recent summary.

- **Outer-Product Rule** (Hebb, 1949; Nakano, 1972; Amari, 1972; Hopfield, 1982). In the attractor neural network case Hopfield (1982), this Hebbian rule constructs weights as the normalized sum of training pattern outer products. This rule is simple, biologically motivated, and local in nature, but it is often observed to suffer from spurious attractors, shallow basins of attraction, and overall limited capacity.

- **Perceptron Rule** (Rosenblatt, 1958). The difference between the desired response – that the network dynamics should fix a training sample – and the actual linear-threshold response of a neuron gives a learning signal to update parameters.

- **Delta** (Hoff & Widrow, 1960; Rescorla & Wagner, 1972). The delta rule, also called the Mean Squared Error (MSE) or Least Mean Square (LMS) rule, considers a relaxation and follows a gradient to minimize the squared error between the linear output activations of neurons and the training pattern to memorize.

- **Projection Rule** (Personnaz et al., 1985; 1986). The weight matrix is obtained by projecting onto the span of the training data and then zeroing the diagonal entries.

- **Storkey Rule** (Storkey, 1997; Storkey & Valabregue, 1999). A modification of the Hebbian update that reduces interference between patterns by accounting for previously stored ones. This rule achieves higher storage capacity than Hebbian learning and reduces spurious minima.

### A.3  Encoding simple, undirected graphs as binary vectors

First we recap some of our notation. Let $\mathcal{G}_v$ denote the the set of all simple, undirected graphs on $v \in \mathbb{N}$ vertices. Recall two graphs $G = (V, E)$, $G' = (V', E')$ are isomorphic, which we denote $G \cong G'$, if there exists a bijection $\phi : V \to V'$ such that $(\phi(\nu_1), \phi(\nu_2)) \in E'$ if and only if $(\nu_1, \nu_2) \in E$. We refer to such a $\phi$ as an isomorphism between $G$ and $G'$. Furthermore, $\phi$ is a permutation when $V = V'$: in our setting we consider $V = V' = [v]$ and therefore we shall discuss only permutations moving forward. The *isomorphism class* of a graph $G \in \mathcal{G}_v$ is defined as $\mathcal{I}(G) := \{G' \in \mathcal{G}_v : G' \cong G\}$. An automorphism of a graph $G = (V, E)$ is a permutation $\phi : V \to V$ such that $(\nu_1, \nu_2) \in E$ implies $(\phi(\nu_1), \phi(\nu_2)) \in E$. In short, while an isomorphism preserves the vertex adjacency structure of a graph an automorphism preserves not just the vertex adjacency structure but also the vertex labels. Recall that $\Phi_n \subset \mathcal{P}_n$ refers to the set of edge permutation matrices induced by permutations of the vertices, see Definition 4.2.

**Lemma A.2.** $\Phi_n$ *is a subgroup of* $\mathcal{P}_n$.

*Proof.* Clearly this is equivalent to showing that $\Pi_n^\Phi$ is a subgroup of $\Pi_n$. It is easy to check that $I \in \Pi_n^\Phi$, $\pi_\phi \in \Pi_n^\Phi$ implies $\pi_\phi^{-1} \in \Pi_n^\Phi$ and $\pi_\phi, \pi_{\phi'} \in \Pi_n^\Phi$ implies $\pi_\phi \circ \pi_{\phi'} \in \Pi_n^\Phi$, therefore $\Pi_n^\Phi$ is a subgroup of $\Pi_n$. $\qquad\square$

The following lemmas establish a straightforward equivalence between isomorphism classes of graphs and certain orbits of binary vectors. First, Lemma A.3 shows that if two graphs $G, G' \in \mathcal{G}_v$ are isomorphic then there is a vertex induced edge permutation which maps between their edge representations.

**Lemma A.3.** *Suppose* $G = ([v], E), G' = ([v], E') \in \mathcal{G}_v$ *and* $\boldsymbol{x} = \mathcal{E}_{rep}(G)$, $\boldsymbol{x}' = \mathcal{E}_{rep}(G')$. *Then* $G \cong G'$ *iff there exists a* $\boldsymbol{P} \in \Phi_n$ *such that* $\boldsymbol{P}\boldsymbol{x} = \boldsymbol{x}'$.

*Proof.* Assume $G \cong G'$. Then there exists a permutation $\phi : [v] \to [v]$ such that $(\nu_1, \nu_2) \in E$ implies $(\phi(\nu_1), \phi(\nu_2)) \in E'$. Let $\pi_\phi : [n] \to [n]$ be the edge permutation induced by $\phi$ and $\boldsymbol{P} \in \Phi_n$ the corresponding permutation matrix. By construction $x_j = x'_{\pi_\phi(j)}$ for all $j \in [n]$, equivalently, if $\boldsymbol{P} \in \Phi_n$ is the permutation matrix associated with $\pi_\phi$ then $\boldsymbol{P}\boldsymbol{x} = \boldsymbol{x}'$. Now suppose there exists a $\boldsymbol{P} \in \Phi_n$ such that $\boldsymbol{P}\boldsymbol{x} = \boldsymbol{x}'$. Then there exists a vertex permutation $\phi : [v] \to [v]$ which induces an edge permutation $\pi_\phi : [n] \to [n]$ such that $x_j = x'_{\pi_\phi(j)}$. By construction, if $j = \mathrm{Ind}((\nu_1, \nu_2))$ then this implies $\pi_\phi(j) = \mathrm{Ind}((\phi(\nu_1), \phi(\nu_2)))$. Therefore, by the definition of $\mathcal{E}_{rep}$ we have $\phi(\nu_1), \phi(\nu_2)) \in E'$ iff $(\nu_1, \nu_2) \in E$. Therefore $\phi$ is an isomorphism between $G$ and $G'$ and $G \cong G'$. $\qquad\square$

Building on Lemma A.3, the following lemma characterizes the isomorphism class of a graph in terms of the orbit of its edge representation under vertex induced edge permutations.

**Lemma A.4.** *For any* $G \in \mathcal{G}_v$ *we have* $\mathcal{E}_{rep}(\mathcal{I}(G)) = \mathrm{Orb}(\mathcal{E}_{rep}(G), \Phi_n)$.

*Proof.* Let $\boldsymbol{x} = \mathcal{E}_{rep}(G)$, then

$$\mathrm{Orb}(\mathcal{E}_{rep}(G), \Phi_n) = \{\boldsymbol{P}\boldsymbol{x} : \boldsymbol{P} \in \Phi_n\}.$$

Suppose $\mathcal{E}_{rep}(\mathcal{I}(G)) \not\subset \mathrm{Orb}(\mathcal{E}_{rep}(G), \Phi_n)$. Then there exists a $G' \in \mathcal{I}(G)$ such that $\boldsymbol{x}' := \mathcal{E}_{rep}(G') \notin \mathrm{Orb}(\mathcal{E}_{rep}(G), \Phi_n)$. Therefore there does not exist a $\boldsymbol{P} \in \Phi_n$ such that $\boldsymbol{P}\boldsymbol{x} = \boldsymbol{x}'$. However, $G \cong G'$ which implies a contradiction by Lemma A.3, therefore $\mathcal{E}_{rep}(\mathcal{I}(G)) \subset \mathrm{Orb}(\mathcal{E}_{rep}(G), \Phi_n)$. Now suppose $\mathrm{Orb}(\mathcal{E}_{rep}(G), \Phi_n) \not\subset \mathcal{E}_{rep}(\mathcal{I}(G))$, then there exists a $\boldsymbol{x}' \in \mathrm{Orb}(\mathcal{E}_{rep}(G), \Phi_n)$ such that $G' = \mathcal{E}_{rep}^{-1}(\boldsymbol{x}') \notin \mathcal{I}(G)$. However, as $\boldsymbol{x}' \in \mathrm{Orb}(\mathcal{E}_{rep}(G), \Phi_n)$ then there exists a $\boldsymbol{P} \in \Phi_n$ such that $\boldsymbol{P}\boldsymbol{x} = \boldsymbol{x}'$, but using Lemma A.3 this implies $G \cong G'$ which is a contradiction. Therefore $\mathrm{Orb}(\mathcal{E}_{rep}(G), \Phi_n) \subset \mathcal{E}_{rep}(\mathcal{I}(G))$. Combining these two observations we conclude that $\mathcal{E}_{rep}(\mathcal{I}(G)) = \mathrm{Orb}(\mathcal{E}_{rep}(G), \Phi_n)$. $\qquad\square$

**Lemma A.5.** $\boldsymbol{Q}_n$ *is a subgroup of* $\mathcal{P}_n$.

*Proof.* Trivially it suffices to show that $\Pi_n^{\mathcal{Q}}$ is a subgroup of $\Pi_n$. It is easy to check that $I \in \Pi_n^{\mathcal{Q}}$, $\pi \in \Pi_n^{\mathcal{Q}}$ implies $\pi^{-1} \in \Pi_n^{\mathcal{Q}}$ and $\pi, \pi' \in \Pi_n^{\mathcal{Q}}$ implies $\pi \circ \pi' \in \Pi_n^{\mathcal{Q}}$. Therefore $\Pi_n^{\mathcal{Q}}$ is a subgroup of $\Pi_n$. □

The following lemma states that the vertex induced edge permutations form a subgroup of the edge adjacency preserving subgroup of permutations. As a result, the edge representations of the isomorphism class of a graph are a subset of the orbit of the edge representation of the graph in question under edge adjacency preserving permutations.

**Lemma A.6.** $\Phi_n$ *is a subgroup of* $\mathcal{Q}_n$ *and* $\mathcal{E}_{rep}(\mathcal{I}(G)) \subset \mathrm{Orb}(\mathcal{E}_{rep}(G), \mathcal{Q}_n)$.

*Proof.* Trivially it suffices to show that $\Pi_n^{\phi}$ is a subgroup of $\Pi_n^{\mathcal{Q}}$, we proceed to show that any vertex induced permutation is an edge adjacency preserving permutation. Consider two edge indices $i, j \in [n]$ and let $\nu_1, \nu_2, \nu_3, \nu_4 \in [v]$ be distinct. Suppose $i \sim j$, then without loss of generality let $\mathrm{Ind}^{-1}(i) = (\nu_1, \nu_2)$ and $\mathrm{Ind}^{-1}(j) = (\nu_2, \nu_3)$. Then $\mathrm{Ind}^{-1}(\pi_\phi(i)) = (\phi(\nu_1), \phi(\nu_2))$ and $\mathrm{Ind}^{-1}(\pi_\phi(j)) = (\phi(\nu_2), \phi(\nu_3))$, therefore $i \sim j$ implies $\pi_\phi(i) \sim \pi_\phi(j)$. Suppose now $i \nsim j$, if $i = j$ then trivially $\pi_\phi(i) = \pi_\phi(j)$ and therefore $i = j$ implies $\pi_\phi(i) \nsim \pi_\phi(i)$. Otherwise, and again without loss of generality, let $\mathrm{Ind}^{-1}(i) = (\nu_1, \nu_2)$ and $\mathrm{Ind}^{-1}(j) = (\nu_3, \nu_4)$. Then $\mathrm{Ind}^{-1}(\pi_\phi(i)) = (\phi(\nu_1), \phi(\nu_2))$ and $\mathrm{Ind}^{-1}(\pi_\phi(j)) = (\phi(\nu_3), \phi(\nu_4))$, as $\phi$ is bijection then this implies $\pi_\phi(i) \nsim \pi_\phi(j)$. As a result, $\pi_\phi(i) \sim \pi_\phi(i)$ if and only if $i \sim j$. Finally as $\Phi_n$ is a group and it is a subset of $\mathcal{Q}_n$ the it must be a subgroup of $\mathcal{Q}_n$. As a result $\mathrm{Orb}(\mathcal{E}_{rep}(G), \Phi_n) \subset \mathrm{Orb}(\mathcal{E}_{rep}(G), \mathcal{Q}_n)$ □

## A.4 Bounded representations

In order to establish the connection between Hopfield networks and SVMs discussed in Section 3, we identified and defined a certain feature map for the inputs to the underlying linear classification problem. Recall there exists a matrix $\boldsymbol{V} \in \{0, 1, 1/\sqrt{2}\}^{n(n+1) \times q}$ such that for any $\boldsymbol{\theta} \in \Theta$ there exists a $\boldsymbol{a} \in \mathbb{R}^p$, $\boldsymbol{\omega} = [\sqrt{2}\boldsymbol{a}, \boldsymbol{b}]$ such that $\boldsymbol{\theta} = \boldsymbol{V}\boldsymbol{\omega}$. Recall also that we define $\boldsymbol{V}_j \in \{0, 1, 1/\sqrt{2}\}^n$ as the matrix which satisfies $\boldsymbol{\theta}_j = \boldsymbol{V}_j \boldsymbol{\omega}$, where $\boldsymbol{\theta}_j = [\boldsymbol{w}_j, b_j] \in \mathbb{R}^{n+1}$. In addition, for any $\boldsymbol{x} \in \{0, 1\}$ then we let $\boldsymbol{z}(\boldsymbol{x}) = [\boldsymbol{x}, 1] \in \mathbb{R}^{n+1}$, $\boldsymbol{u}_j(\boldsymbol{x}) = \boldsymbol{V}_j^T \boldsymbol{z}(\boldsymbol{x})$ for all $j \in [n]$ and $\bar{\boldsymbol{u}}(\boldsymbol{x}) = \frac{1}{n} \sum_{j=1}^n \boldsymbol{u}_j(\boldsymbol{x})$. The following lemma bounds the norm of these representations.

**Lemma A.7.** *For any* $\boldsymbol{x} \in \{0, 1\}^n$ *then*

$$\|\bar{\boldsymbol{u}}(\boldsymbol{x})\|^2 \leq \|\boldsymbol{u}_j(\boldsymbol{x})\|^2 = \frac{1}{2} \left( \|\boldsymbol{x}\|_0 - x_j \right) + 1$$

*for all* $j \in [n]$.

*Proof.* Let $\delta_n(j) \in \{0, 1\}^n$ denote the one hot vector such that $\mathrm{supp}(\delta_n(j)) = j$. In addition, let $\phi_j : [q] \to [n]$ denote the injective mapping between the indices of $\boldsymbol{a}$ and their respective positions in $\boldsymbol{w}_j$, and let $\boldsymbol{B}_j \in \{0, 1\}^{n \times p}$ be the associated matrix which copies the elements of $\boldsymbol{a}$ into their positions in $\boldsymbol{\theta}_j$. Therefore, we can write

$$\boldsymbol{\theta}_j = \begin{bmatrix} \boldsymbol{w}_j \\ b_j \end{bmatrix} = \begin{bmatrix} \frac{1}{\sqrt{2}}\boldsymbol{B}_j & \boldsymbol{0}_{n \times n} \\ \boldsymbol{0}_{1 \times n} & \delta_n(j)^T \end{bmatrix} \begin{bmatrix} \sqrt{2}\boldsymbol{a} \\ \boldsymbol{b} \end{bmatrix} = \boldsymbol{V}_j \boldsymbol{\omega}.$$

By definition

$$\boldsymbol{u}_j(\boldsymbol{x}) = \boldsymbol{V}_j^T \boldsymbol{z}(\boldsymbol{x}) = \begin{bmatrix} \frac{1}{\sqrt{2}}\boldsymbol{B}_j^T & \boldsymbol{0}_{n \times 1} \\ \boldsymbol{0}_{n \times n} & \delta_n(j) \end{bmatrix} \begin{bmatrix} \boldsymbol{x} \\ 1 \end{bmatrix} = \begin{bmatrix} \frac{1}{\sqrt{2}}\boldsymbol{B}_j^T \boldsymbol{x} \\ \delta_n(j) \end{bmatrix},$$

therefore

$$\|\boldsymbol{u}_j(\boldsymbol{x})\|^2 = \frac{1}{2}\boldsymbol{x}^T \boldsymbol{B}_j \boldsymbol{B}_j^T \boldsymbol{x} + 1.$$

Recall $W_{jj} = 0$ and each other element of $\boldsymbol{w}_j$ corresponds to exactly one element of $\boldsymbol{a}$, therefore $\boldsymbol{B}_j$ has one nonzero per row other than the $j$th row, which we let be zero. Moreover, by the injectivity of $\phi_j$ then $\boldsymbol{B}_j$ has at most one nonzero per column. As a result,

$$\boldsymbol{B}_j \boldsymbol{B}_j^T = \mathbf{I}_n - \boldsymbol{e}_j \boldsymbol{e}_j^T.$$

This implies

$$\boldsymbol{x}^T \boldsymbol{B}_j \boldsymbol{B}_j^T \boldsymbol{x} = \boldsymbol{x}^T \left( \mathbf{I}_n - \boldsymbol{e}_j \boldsymbol{e}_j^T \right) \boldsymbol{x} = \|\boldsymbol{x}\|_0 - x_j$$

for all $j \in [n]$. As

$$\|\bar{\boldsymbol{u}}(\boldsymbol{x})\| = \|\frac{1}{n} \sum_{j=1}^n \boldsymbol{u}_j(\boldsymbol{x})\| \le \frac{1}{n} \sum_{j=1}^n \|\boldsymbol{u}_j(\boldsymbol{x})\| \le \|\boldsymbol{u}_j(\boldsymbol{x}),\|$$

then

$$\|\bar{\boldsymbol{u}}(\boldsymbol{x})\|^2 \le \|\boldsymbol{u}_j(\boldsymbol{x})\|^2 = \frac{1}{2} \left( \|\boldsymbol{x}\|_0 - x_j \right) + 1$$

for all $j \in [n]$ as claimed. $\qquad\square$

## A.5   Euclidean distance bounds between normalized vectors

Here we recall some basic bounds pertaining to normalized vectors.

**Lemma A.8.** *Define* $f(\boldsymbol{x}) = \frac{\boldsymbol{x}}{\|\boldsymbol{x}\|^2}$ *for all* $\boldsymbol{x} \in \mathbb{R}^q$. *Suppose without loss of generality that* $\boldsymbol{x}, \boldsymbol{y} \in \mathbb{R}^q$ *and* $\|\boldsymbol{y}\| \ge \|\boldsymbol{x}\| > 0$, *then*

$$\|f(\boldsymbol{x}) - f(\boldsymbol{y})\| \le \frac{3\|\boldsymbol{x} - \boldsymbol{y}\|}{\|\boldsymbol{x}\|^2}.$$

*Proof.* First observe

$$\begin{aligned}
f(\boldsymbol{x}) - f(\boldsymbol{y}) &= \frac{\boldsymbol{x}}{\|\boldsymbol{x}\|^2} - \frac{\boldsymbol{y}}{\|\boldsymbol{y}\|^2} \\
&= \frac{\boldsymbol{x}}{\|\boldsymbol{x}\|^2} - \frac{\boldsymbol{y}}{\|\boldsymbol{y}\|^2} + \frac{\boldsymbol{y}}{\|\boldsymbol{x}\|^2} - \frac{\boldsymbol{y}}{\|\boldsymbol{x}\|^2} \\
&= \frac{\boldsymbol{x} - \boldsymbol{y}}{\|\boldsymbol{x}\|^2} + \frac{\boldsymbol{y}(\|\boldsymbol{y}\|^2 - \|\boldsymbol{x}\|^2)}{\|\boldsymbol{x}\|^2 \|\boldsymbol{y}\|^2}.
\end{aligned}$$

Taking the norm on both sides and applying the triangle inequality we have

$$\|f(\boldsymbol{x}) - f(\boldsymbol{y})\| = \frac{\|\boldsymbol{x} - \boldsymbol{y}\|}{\|\boldsymbol{x}\|^2} + \frac{\|\boldsymbol{y}\|(|\|\boldsymbol{y}\|^2 - \|\boldsymbol{x}\|^2|)}{\|\boldsymbol{x}\|^2 \|\boldsymbol{y}\|^2}.$$

By assumption $\|\boldsymbol{y}\| \ge \|\boldsymbol{x}\|$, therefore

$$|\|\boldsymbol{y}\|^2 - \|\boldsymbol{x}\|^2| = |\langle \boldsymbol{y} - \boldsymbol{x}, \boldsymbol{y} + \boldsymbol{x}\rangle| \le \|\boldsymbol{y} - \boldsymbol{x}\|\|\boldsymbol{y} + \boldsymbol{x}\| \le \|\boldsymbol{y} - \boldsymbol{x}\|(\|\boldsymbol{y}\| + \|\boldsymbol{x}\|) \le 2\|\boldsymbol{y}\|\|\boldsymbol{y} - \boldsymbol{x}\|.$$

This implies

$$\begin{aligned}
\|f(\boldsymbol{x}) - f(\boldsymbol{y})\| &\le \frac{\|\boldsymbol{x} - \boldsymbol{y}\|}{\|\boldsymbol{x}\|^2} + \frac{2\|\boldsymbol{y}\|^2 \|\boldsymbol{y} - \boldsymbol{x}\|}{\|\boldsymbol{x}\|^2 \|\boldsymbol{y}\|^2} \\
&= \frac{3\|\boldsymbol{x} - \boldsymbol{y}\|}{\|\boldsymbol{x}\|^2}
\end{aligned}$$

as claimed. $\qquad\square$

## A.6   Hoeffding's inequality in Hilbert space

Lemma 4.10 rests on the application of the following concentration bound for sums of independent, mean zero, bounded random vectors. We remark that this is a specialization of more general results for martingales in 2-smooth Banach spaces.

**Lemma A.9.** *[Specialization of (Pinelis, 1994, Thm. 3.5)* For $i \in [N]$ and $R \in \mathbb{R}_{>0}$, let $\boldsymbol{x}_i \in \mathbb{R}^q$ be independent, mean zero random vectors which satisfy $\|\boldsymbol{x}_i\| \le R$ almost surely. Let $S_N = \sum_{i=1}^N \boldsymbol{x}_i$, then for $t \in \mathbb{R}_{\ge 0}$ we have

$$\mathbb{P}(\|S_N\| \ge t) \le \exp\left( -\frac{t^2}{2NR^2} \right).$$

# B  Proofs of results

## B.1  Proof of Theorem 3.2

**Theorem 3.2.** *Let $D \subset \{0,1\}^n$ be a set which can be strictly memorized and assume $\|\boldsymbol{x}\|_0 \leq m \in \mathbb{N}_{\geq 1}$ for all $\boldsymbol{x} \in D$. Let $\mathcal{D}$ be a probability distribution on $D$, and consider a random sample $\mathcal{S}_N = (\boldsymbol{x}_i)_{i \in [N]}$, where $\boldsymbol{x}_i \sim \mathcal{D}$ are mutually i.i.d. Let $\hat{\boldsymbol{\omega}} = \mathrm{HSVM}(\mathcal{S}_N)$, $\boldsymbol{\omega}^* = \mathrm{HSVM}(D)$, $\boldsymbol{\omega}(0) \in \mathbb{R}^q$ be arbitrary and $\boldsymbol{\omega}(t)$ be generated for all $t \in \mathbb{N}_{\geq 1}$ as per (8), $\delta, \epsilon \in (0,1)$ and assume $\boldsymbol{x} \sim \mathcal{D}$ is sampled independent of $\mathcal{S}_N$. If $N \gtrsim \epsilon^{-2} n \|\boldsymbol{\omega}^*\|^2 m \log(1/\delta)$ then both*

$$\mathbb{P}(H(\boldsymbol{x}; \boldsymbol{V}\hat{\boldsymbol{\omega}}) \neq \boldsymbol{x}) \leq \epsilon \quad and \quad \mathbb{P}(H(\boldsymbol{x}; \boldsymbol{V}\boldsymbol{\omega}(t)) \neq \boldsymbol{x}) = \tilde{O}\left(\frac{\sqrt{m}\|\boldsymbol{\omega}^*\|}{\log(t)}\right) + \epsilon$$

*hold with probability at least $1 - \delta$ over the sample $\mathcal{S}_N$.*

*Proof.* For any $t \in \mathbb{R}$ define the margin loss $\phi : \mathbb{R} \to \mathbb{R}$ as

$$\phi(t) = \begin{cases} 0, & 1 \leq t, \\ 1 - t, & 0 \leq t \leq 1, \\ 1, & t \leq 0, \end{cases}$$

and note trivially for all $t \in \mathbb{R}$ that $\mathbb{1}(t \leq 0) \leq \phi(t)$. We note on occasion we overload this notation and apply $\phi$ to vectors by applying it elementwise. Observe for any $\boldsymbol{x} \in \mathbb{R}^n$ and $\boldsymbol{\omega} \in \mathbb{R}^q$ with $\boldsymbol{\theta} = \boldsymbol{V}\boldsymbol{\omega}$, that

$$\mathbb{1}(H(\boldsymbol{x}; \boldsymbol{\theta}) \neq \boldsymbol{x}) \leq \mathbb{1}\left(\exists j \in [n] : \boldsymbol{u}_j(\boldsymbol{x})^T \omega \leq 0\right) = \mathbb{1}\left(\min_{j \in [n]} \boldsymbol{u}_j(\boldsymbol{x})^T \boldsymbol{\omega} \leq 0\right) \leq \phi(\min_{j \in [n]} \boldsymbol{u}_j(\boldsymbol{x})^T \boldsymbol{\omega}) = \max_{j \in [n]} \phi(\boldsymbol{u}_j(\boldsymbol{x})^T \boldsymbol{\omega}).$$

For any $\boldsymbol{z} \in \mathbb{R}^n$, let $\ell(\boldsymbol{z}) = \max_{j \in [n]} \phi(z_j)$. Using the fact that $\phi$ is 1-Lipschitz, for any $\boldsymbol{z}, \boldsymbol{z}' \in \mathbb{R}^n$ we have

$$|\ell(\boldsymbol{z}) - \ell(\boldsymbol{z}')| \leq \max_{j \in [n]} |\phi(z_j) - \phi(z_j')| = \|\phi(\boldsymbol{z}) - \phi(\boldsymbol{z}')\|_\infty \leq \|\boldsymbol{z} - \boldsymbol{z}'\|_\infty \leq \|\boldsymbol{z} - \boldsymbol{z}'\|_2.$$

Therefore $\ell$ is 1-Lipschitz with respect to the Euclidean norm. Let $\boldsymbol{U}(\boldsymbol{x}) \in \mathbb{R}^{n \times q}$ denote the matrix whose $j$th row is $\boldsymbol{u}_j(\boldsymbol{x})^T \in \mathbb{R}^{1 \times q}$ for all $j \in [n]$. Furthermore, for some $\Lambda \in \mathbb{R}_{>0}$, define

$$\mathcal{H}_\Lambda = \{\boldsymbol{x} \mapsto \boldsymbol{U}(\boldsymbol{x})\boldsymbol{\omega} : \boldsymbol{\omega} \in \mathbb{R}^q, \|\boldsymbol{\omega}\| \leq \Lambda\}$$

and let

$$\mathcal{G}_\Lambda = \{\boldsymbol{x} \mapsto (\ell \circ h)(\boldsymbol{x}) : h \in \mathcal{H}_\Lambda\}.$$

Note by construction that $g \in \mathcal{G}_\Lambda$ implies $g : \mathbb{R}^n \to [0,1]$. We now compute the empirical Rademacher complexity of $\mathcal{G}_\Lambda$ on a sample $\mathcal{S}_N = (\boldsymbol{x}_i)_{i \in [N]}$, to this end let $\boldsymbol{\sigma} \in \{\pm 1\}^N$ and $\boldsymbol{\epsilon} \in \{\pm\}^{N \times n}$ be a random vector and matrix respectively whose entries are mutually i.i.d. and distributed uniformly on $\{\pm 1\}$. As $\ell$ is 1-Lipschitz in the $\ell_2$ norm, then applying a vector contraction inequality (Maurer, 2016, Corollary 1) we have

$$\tilde{\mathfrak{R}}_\mathcal{S}(\mathcal{G}_\Lambda) = \mathbb{E}_{\boldsymbol{\sigma}}\left[\sup_{h \in \mathcal{H}_\Lambda} \frac{1}{N} \sum_{i=1}^N \sigma_i (\ell \circ h)(\boldsymbol{x}_i)\right]$$

$$\leq \sqrt{2} E_{\boldsymbol{\epsilon}}\left[\sup_{\|\boldsymbol{\omega}\| \leq \Lambda} \frac{1}{N} \sum_{i=1}^N \sum_{j=1}^n \epsilon_{ij} \boldsymbol{u}_j(\boldsymbol{x}_i)^T \boldsymbol{\omega}\right]$$

$$\leq \sqrt{2} \mathbb{E}_{\boldsymbol{\epsilon}}\left[\sup_{\|\boldsymbol{\omega}\| \leq \Lambda} \langle \boldsymbol{\omega}, \frac{1}{N} \sum_{i=1}^N \sum_{j=1}^n \epsilon_{ij} \boldsymbol{u}_j(\boldsymbol{x}_i) \rangle\right]$$

$$\leq \frac{\sqrt{2}\Lambda}{N} \mathbb{E}_{\boldsymbol{\epsilon}}\left[\left\|\sum_{i=1}^N \sum_{j=1}^n \epsilon_{ij} \boldsymbol{u}_j(\boldsymbol{x}_i)\right\|\right].$$

Let $Z = \sum_{i=1}^{N} \sum_{j=1}^{n} \epsilon_{ij} \boldsymbol{u}_j(\boldsymbol{x}_i)$, as $t \mapsto \sqrt{t}$ is concave then $\mathbb{E}_Z[\sqrt{\|Z\|^2}] \leq \sqrt{\mathbb{E}_Z[\|Z\|^2]}$ by Jensen's inequality. As a result

$$\mathbb{E}_{\boldsymbol{\epsilon}} \left[ \left\| \sum_{i=1}^{N} \sum_{j=1}^{n} \epsilon_{ij} \boldsymbol{u}_j(\boldsymbol{x}_i) \right\| \right] \leq \sqrt{\mathbb{E}_{\boldsymbol{\epsilon}} \left[ \left\langle \sum_{i=1}^{N} \sum_{j=1}^{n} \epsilon_{ij} \boldsymbol{u}_j(\boldsymbol{x}_i), \sum_{l=1}^{N} \sum_{k=1}^{n} \epsilon_{lk} \boldsymbol{u}_k(\boldsymbol{x}_l) \right\rangle \right]}$$

$$= \sqrt{\sum_{i=1}^{N} \sum_{j=1}^{n} \sum_{l=1}^{N} \sum_{k=1}^{n} \mathbb{E}_{\boldsymbol{\epsilon}}[\epsilon_{ij}\epsilon_{lk}] \boldsymbol{u}_j(\boldsymbol{x}_i)^T \boldsymbol{u}_k(\boldsymbol{x}_l)}.$$

The Rademacher random variables are mutually i.i.d., therefore

$$\mathbb{E}_{\boldsymbol{\epsilon}}[\epsilon_{ij}\epsilon_{lk}] = \begin{cases} 1, & (i = l) \wedge (j = k), \\ 0, & \text{otherwise.} \end{cases}$$

Recall also from Lemma A.7 that for any $i \in [N]$ and for all $j \in [n]$

$$\|\boldsymbol{u}_j(\boldsymbol{x}_i)\|^2 = \frac{1}{2} \left( \|\boldsymbol{x}_i\|_0 - x_{ij} \right) + 1.$$

Under the assumption $\|\boldsymbol{x}_i\|_0 \leq m$ for all $i \in [N]$, then

$$\tilde{\mathfrak{R}}_{\mathcal{S}}(\mathcal{G}_\Lambda) \leq \frac{\sqrt{2}\Lambda}{N} \mathbb{E}_{\boldsymbol{\epsilon}} \left[ \left\| \sum_{i=1}^{N} \sum_{j=1}^{n} \epsilon_{ij} \boldsymbol{u}_j(\boldsymbol{x}_i) \right\| \right]$$

$$\leq \frac{\sqrt{2}\Lambda}{N} \sqrt{\sum_{i=1}^{N} \sum_{j=1}^{n} \|\boldsymbol{u}_j(\boldsymbol{x}_i)\|^2}$$

$$\leq \frac{\Lambda}{N} \sqrt{\sum_{i=1}^{N} \sum_{j=1}^{n} (\|\boldsymbol{x}_i\|_0 - x_{ij} + 2)}$$

$$\leq \Lambda \sqrt{\frac{n(m+2)}{N}}.$$

Let $\delta \in \mathbb{R}_{>0}$. Applying a Rademacher complexity bound, e.g., (Mohri et al., 2018, Thm 3.3), then with probability at least $1 - \delta$ over the random sample $\mathcal{S}_N$

$$\mathbb{E}[g(\boldsymbol{x})] \leq \frac{1}{N} \sum_{i=1}^{N} g(\boldsymbol{x}_i) + 2\tilde{\mathfrak{R}}_{\mathcal{S}}(\mathcal{G}_\Lambda) + 3\sqrt{\frac{\log(2/\delta)}{2N}}$$

$$\leq \frac{1}{N} \sum_{i=1}^{N} g(\boldsymbol{x}_i) + 2\Lambda \sqrt{\frac{n(m+2)}{N}} + 3\sqrt{\frac{\log(2/\delta)}{2N}}$$

for all $g \in \mathcal{G}_\Lambda$. In what follows let $\Lambda = \|\boldsymbol{\omega}^*\|$. Then, with probability at least $1 - \delta$ over the random sample $\mathcal{S}_N$, for any $\boldsymbol{\omega} \in \mathbb{R}^q$ such that $\|\boldsymbol{\omega}\| \leq \|\boldsymbol{\omega}^*\|$ we have

$$\mathbb{P}(H(\boldsymbol{x}; \boldsymbol{V}\boldsymbol{\omega}) \neq \boldsymbol{x}) \leq \frac{1}{N} \sum_{i=1}^{N} \phi(\min_{j \in [n]} \boldsymbol{u}_j(\boldsymbol{x}_i)^T \boldsymbol{\omega})) + 2\sqrt{\frac{\|\boldsymbol{\omega}^*\|^2 n(m+2)}{N}} + 3\sqrt{\frac{\log(2/\delta)}{2N}} \tag{10}$$

First we consider $\hat{\boldsymbol{\omega}}$: for any sample $\mathcal{S}_N$ trivially set$(\mathcal{S}_N) \subseteq D$, therefore $\boldsymbol{\omega}^* \in \mathcal{F}_{\boldsymbol{\omega}}(\mathcal{S}_N)$. As a result, with probability one $\|\hat{\boldsymbol{\omega}}\| \leq \|\boldsymbol{\omega}^*\|$. Conditioning on this event, observe also that $\min_{j \in [n]} \boldsymbol{u}_j(\boldsymbol{x}_i)^T \hat{\boldsymbol{\omega}} \geq 1$ for $i \in [N]$. As a result, with probability at least $1 - \delta$ over $\mathcal{S}_N$ we have

$$\mathbb{P}(H(\boldsymbol{x}; \boldsymbol{V}\hat{\boldsymbol{\omega}}) \neq \boldsymbol{x}) \leq \sqrt{\frac{4\|\boldsymbol{\omega}^*\|^2 n(m+2)}{N}} + \sqrt{\frac{9\log(2/\delta)}{2N}}.$$

As $(a+b)^2 \leq 2(a^2 + b^2)$ for $a, b \in \mathbb{R}$ and assuming $m \geq 1$, then for any $\epsilon \in \mathbb{R}_{>0}$, if $N \gtrsim \epsilon^{-2} n \|\boldsymbol{\omega}^*\|^2 m \log(1/\delta)$

$$\mathbb{P}_{\boldsymbol{x}}(H(\boldsymbol{x}; \hat{\boldsymbol{\theta}}) \neq \boldsymbol{x}) \leq \epsilon$$

with probability at least $1 - \delta$ over the sample $\mathcal{S}_N$.

We now turn our attention to $\boldsymbol{\omega}(t)$: recall for any $a \in \mathbb{R}_{>0}$ as $E(\boldsymbol{x}; a\boldsymbol{\theta}) = aE(\boldsymbol{x}; \boldsymbol{\theta})$ then

$$a(E(\boldsymbol{x}^{(j)}; \boldsymbol{\theta}) - E(\boldsymbol{x}; \boldsymbol{\theta})) \geq 0 \iff E(\boldsymbol{x}^{(j)}; a\boldsymbol{\theta}) - E(\boldsymbol{x}; a\boldsymbol{\theta}) \geq 0.$$

Indeed, this implies the set of memories in a Hopfield network is invariant under positive re-scalings of the parameters. As a consequence, for any distribution $\mathcal{D}$ on $\{0, 1\}^n$, $a \in \mathbb{R}_{>0}$ and $\boldsymbol{\theta} \in \Omega$, if $\boldsymbol{x} \sim \mathcal{D}$ we have

$$\mathbb{P}(H(\boldsymbol{x}; \boldsymbol{\theta}) \neq \boldsymbol{x}) = \mathbb{P}(H(\boldsymbol{x}; a\boldsymbol{\theta}) \neq \boldsymbol{x}).$$

Therefore, if we define

$$\bar{\boldsymbol{\omega}}(t) = \frac{\|\hat{\boldsymbol{\omega}}\|}{\|\boldsymbol{\omega}(t)\|} \boldsymbol{\omega}(t)$$

it follows that

$$\mathbb{P}(H(\boldsymbol{x}; \boldsymbol{V}\boldsymbol{\omega}(t)) \neq \boldsymbol{x}) = \mathbb{P}(H(\boldsymbol{x}; \boldsymbol{V}\bar{\boldsymbol{\omega}}(t)) \neq \boldsymbol{x}).$$

From (Soudry et al., 2018, Theorem 5),

$$\left\| \frac{\bar{\boldsymbol{\omega}}(t)}{\|\hat{\boldsymbol{\omega}}\|} - \frac{\hat{\boldsymbol{\omega}}}{\|\hat{\boldsymbol{\omega}}\|} \right\| = \left\| \frac{\boldsymbol{\omega}(t)}{\|\boldsymbol{\omega}(t)\|} - \frac{\hat{\boldsymbol{\omega}}}{\|\hat{\boldsymbol{\omega}}\|} \right\| = O\left( \frac{\log(\log(t))}{\log(t)} \right).$$

Recalling $\|\hat{\boldsymbol{\omega}}\| \leq \|\boldsymbol{\omega}^*\|$ this implies

$$\|\bar{\boldsymbol{\omega}}(t) - \hat{\boldsymbol{\omega}}\| = O\left( \frac{\|\boldsymbol{\omega}^*\| \log(\log(t))}{\log(t)} \right).$$

As a result, for all $i \in [N]$ we have

$$\begin{aligned}
\min_{j \in [n]} \boldsymbol{u}_j(\boldsymbol{x}_i)^T \bar{\boldsymbol{\omega}}(t) &= \min_{j \in [n]} \left( \boldsymbol{u}_j(\boldsymbol{x}_i)^T \hat{\boldsymbol{\omega}} + \boldsymbol{u}_j(\boldsymbol{x}_i)^T (\bar{\boldsymbol{\omega}}(t) - \hat{\boldsymbol{\omega}}) \right) \\
&\geq 1 - \max_{j \in [n]} \|\boldsymbol{u}_j(\boldsymbol{x}_i)\| \|\bar{\boldsymbol{\omega}}(t) - \hat{\boldsymbol{\omega}}\| \\
&\geq 1 - O\left( \frac{\sqrt{m} \|\boldsymbol{\omega}^*\| \log(\log(t))}{\log(t)} \right),
\end{aligned}$$

where the final inequality follows from Lemma A.7 and the assumption $m \geq 1$. By the definition of $\phi$ it follows that

$$\phi\left( \min_{j \in [n]} \boldsymbol{u}_j(\boldsymbol{x}_i)^T \bar{\boldsymbol{\omega}}(t) \right) = O\left( \frac{\sqrt{m} \|\boldsymbol{\omega}^*\| \log(\log(t))}{\log(t)} \right)$$

for all $i \in [N]$. Using (10), this implies with probability at least $1 - \delta$ over $\mathcal{S}_N$

$$\mathbb{P}(H(\boldsymbol{x}; \boldsymbol{V}\boldsymbol{\omega}(t)) \neq \boldsymbol{x}) \leq O\left( \frac{\sqrt{m} \|\boldsymbol{\omega}^*\| \log(\log(t))}{\log(t)} \right) + \sqrt{\frac{4\|\boldsymbol{\omega}^*\|^2 nm}{N}} + \sqrt{\frac{9 \log(2/\delta)}{N}}.$$

As a result, if $N \gtrsim \epsilon^{-2} n \|\boldsymbol{\omega}^*\|^2 m \log(1/\delta)$ then

$$\mathbb{P}(H(\boldsymbol{x}; \boldsymbol{V}\boldsymbol{\omega}(t)) \neq \boldsymbol{x}) = \tilde{O}\left( \frac{\sqrt{m} \|\boldsymbol{\omega}^*\|}{\log(t)} \right) + \epsilon.$$

with probability at least $1 - \delta$ over the sample $\mathcal{S}_N$. $\qquad \square$

### B.2 Properties of invariant parameters

#### B.2.1 Energy bounds for invariant parameters across orbits

The following result states, in the context of invariant parameters, that energy bound differences for a point in an orbit extend to the entire orbit.

**Lemma 4.5.** *Let $\boldsymbol{x}_0 \in \{0,1\}^n$ and $\boldsymbol{\theta} \in \Psi(\Gamma_n)$. For $\delta \in \mathbb{R}$, if $E(\boldsymbol{x}_0^{(j)}; \boldsymbol{\theta}) - E(\boldsymbol{x}_0; \boldsymbol{\theta}) \geq 1 - \delta$ for all $j \in [n]$, then for all $\boldsymbol{x} \in \mathrm{Orb}(\boldsymbol{x}_0, \Gamma_n)$ it follows that $E(\boldsymbol{x}^{(j)}; \boldsymbol{\theta}) - E(\boldsymbol{x}; \boldsymbol{\theta}) \geq 1 - \delta$ for all $j \in [n]$.*

*Proof.* As $\boldsymbol{\theta} \in \Psi(\Gamma_n)$, recall the intertwining relation (9), for $\boldsymbol{x} \in \{0,1\}^n$ and $\boldsymbol{Q} \in \Gamma_n$ then

$$E(\boldsymbol{Q}\boldsymbol{x}; \boldsymbol{\theta}) = E(\boldsymbol{x}; \boldsymbol{Q}\boldsymbol{\theta}) = E(\boldsymbol{x}; \boldsymbol{\theta}).$$

As for any $\boldsymbol{x} \in \mathrm{Orb}(\boldsymbol{x}_0, \Gamma_n)$ there exists a $\boldsymbol{Q} \in \Gamma_n$, corresponding to a permutation $\pi \in \Pi_n^{\mathcal{Q}}$, such that $\boldsymbol{x}_0 = \boldsymbol{Q}\boldsymbol{x}$, this implies

$$1 - \delta \leq E(\boldsymbol{x}_0^{(j)}; \boldsymbol{\theta}) - E(\boldsymbol{x}_0; \boldsymbol{\theta}) = E((\boldsymbol{Q}\boldsymbol{x})^{(j)}; \boldsymbol{\theta}) - E(\boldsymbol{Q}\boldsymbol{x}; \boldsymbol{\theta}) = E(\boldsymbol{x}^{\pi(j)}; \boldsymbol{\theta}) - E(\boldsymbol{x}; \boldsymbol{\theta}).$$

As $\pi$ is a bijection then for all $j \in [n]$ this implies

$$E(\boldsymbol{x}^{(j)}; \boldsymbol{\theta}) - E(\boldsymbol{x}; \boldsymbol{\theta}) \geq 1 - \delta$$

as claimed. $\qquad\square$

#### B.2.2 Characterizing the invariant set for edge adjacency preserving permutations

The following lemma identifies the invariant parameters with respect to the set of edge adjacency preserving permutations as a particular rank three subspace.

**Lemma 4.6.** *Let $F : \mathbb{R}^3 \to \Theta$ denote the linear map defined as follows: if $(\boldsymbol{W}, \boldsymbol{b}) = F(\boldsymbol{\beta})$ then for all $i, j \in [n]$ we have $w_{ij} = 0$ if $i = j$, $w_{ij} = \beta_1$ if $i \sim j$, $w_{ij} = \beta_2$ if $i \nsim j$ and $b_j = \beta_3$. Then $\Psi(\mathcal{Q}_n) = F(\mathbb{R}^3)$ where $F(\mathbb{R}^3)$ denotes the image of $F$.*

*Proof.* First we show that $F(\mathbb{R}^3) \subseteq \Psi(\mathcal{Q}_n)$. Suppose $\boldsymbol{\theta} \in F(\mathbb{R}^3)$, then there exists $\boldsymbol{\beta} \in \mathbb{R}^3$ such that $\boldsymbol{\theta} = F(\boldsymbol{\beta})$. Let $\boldsymbol{Q} \in \mathcal{Q}_n$ and $\pi \in \Pi_n^Q$ be the corresponding permutation. Then $b_i = b_{\pi(i)} = \beta_3$ for all $i \in [n]$ and as a result $\boldsymbol{Q}^T \boldsymbol{b} = \boldsymbol{b}$. Furthermore, if $i, j \in [n]$ then $\pi(i) \sim \pi(j)$ if and only if $i \sim j$. Therefore if $i \sim j$ then $W_{ij} = \beta_1 = W_{\pi(i)\pi(j)}$. Otherwise, if $i \nsim j$ then either $i = j$, which implies $W_{jj} = 0 = W_{\pi(j)\pi(j)}$, or $i \neq j$ and then $W_{ij} = \beta_2 = W_{\pi(i)\pi(j)}$. As a result it follows that $\boldsymbol{Q}^T \boldsymbol{W} \boldsymbol{Q} = \boldsymbol{W}$, this implies $\boldsymbol{Q}\boldsymbol{\theta} = \boldsymbol{\theta}$ and so $\boldsymbol{\theta} \in \Psi(\mathcal{Q}_n)$. We therefore conclude that $F(\mathbb{R}^3) \subseteq \Psi(\mathcal{Q}_n)$.

Now assume $\boldsymbol{\theta} \in \Psi(\mathcal{Q}_n)$, to prove there exists a $\beta_3 \in \mathbb{R}$ such that $b_i = \beta_3$ for all $i \in [n]$ it suffices to show $b_i = b_j$ for all $i, j \in [n]$. Similarly, to show there exist $\beta_1, \beta_2 \in \mathbb{R}$ as per the statement of the lemma, it suffices to show $w_{ij} = w_{ab}$ whenever either of the following hold: i) $i \sim j$ and $a \sim b$ or ii) $i \nsim j$ and $a \nsim b$. It therefore suffices to prove the following two statements.

1. For any $i, j \in [n]$ there exists a $\pi \in \Pi_n^Q$ such that $\pi(i) = j$. Note, as $\boldsymbol{\theta} \in \Psi(\mathcal{Q}_n)$ this implies $b_i = b_{\pi(i)} = b_j$.

2. For any $i, j, a, b \in [n]$ satisfying $i \sim j$ and $a \sim b$, or $i \nsim j$ and $a \nsim b$, there exists a $\pi \in \Pi_n^Q$ such that $\pi(i) = a$ and $\pi(j) = b$. Note, and again as $\boldsymbol{\theta} \in \Psi(\mathcal{Q}_n)$, this implies $w_{ij} = w_{\pi(i)\pi(j)} = w_{ab}$.

In all that follows, for $l \in [8]$ let $\nu_l \in [v]$. To prove the first statement, let $i, j \in [n]$ and suppose $i = \mathrm{Ind}(\{\nu_1, \nu_2\})$ and $j = \mathrm{Ind}(\{\nu_3, \nu_4\})$. Consider the permutation $\pi \in \Pi_n^{\Phi} \subseteq \Pi_n^Q$ which swaps the indices of the unordered vertex pairs involving $\nu_1$ with the corresponding pair involving $\nu_3$, likewise for $\nu_2$ and $\nu_4$, and is identity otherwise. To be clear, this is the permutation satisfying for $t \in [2]$ the identities $\pi(\mathrm{Ind}(\{\nu_t, \nu\})) = \mathrm{Ind}(\{\nu_{t+2}, \nu\})$ and $\pi(\mathrm{Ind}(\{\nu_{t+2}, \nu\})) = \mathrm{Ind}(\{\nu_t, \nu\})$ for all $\nu \in [v]\setminus\{\nu_t, \nu_{t+2}\}$, and $\pi(\mathrm{Ind}^{-1}(\{\nu, \nu'\})) = \mathrm{Ind}^{-1}(\{\nu, \nu'\})$ for all $\nu, \nu' \in [v]\setminus\{\nu_1, ...\nu_4\}$. Note if $i \sim j$ then we can without loss of generality assume $\nu_2 = \nu_4$ and this

permutation reduces to swapping a single pair and treating the rest with identity. By construction this permutation preserves adjacency, moreover $\pi(i) = j$. As a result, for all $i, j \in [n]$ there exists a $\pi \in \Pi_n^Q$ such that $b_i = b_{\pi(i)} = b_j$.

To prove the second statement, let $i, j, a, b \in [n]$ and suppose $i = \text{Ind}(\{\nu_1, \nu_2\})$, $j = \text{Ind}(\{\nu_3, \nu_4\})$, $a = \text{Ind}(\{\nu_5, \nu_6\})$ and $j = \text{Ind}(\{\nu_7, \nu_8\})$. Now consider the permutation $\pi \in \Pi_n^\Phi \subseteq \Pi_n^Q$ which swaps the indices of the unordered vertex pairs involving $\nu_1$ with those of $\nu_5$, $\nu_2$ with those of $\nu_6$, $\nu_3$ with those of $\nu_7$, $\nu_4$ with those of $\nu_8$ and acts as identity on the indices of all other edges. To be clear, this is the permutation satisfying for $t \in [4]$ the identities $\pi(\text{Ind}^{-1}(\{\nu_t, \nu\})) = \text{Ind}^{-1}(\{\nu_{t+4}, \nu\})$ and $\pi(\text{Ind}^{-1}(\{\nu_{t+4}, \nu\})) = \text{Ind}^{-1}(\{\nu_t, \nu\})$ for all $\nu \in [v] \setminus \{\nu_t, \nu_{t+4}\}$, and $\pi(\text{Ind}^{-1}(\{\nu, \nu'\})) = \text{Ind}^{-1}(\{\nu, \nu'\})$ for all $\nu, \nu' \in [v] \setminus \{\nu_1, ... \nu_8\}$. By construction this permutation preserves adjacency and $\pi(i) = a$, $\pi(j) = b$. Moreover as $\pi \in \Pi_n^Q$ then $i \sim j$ implies $a \sim b$ and $i \not\sim j$ implies $a \not\sim b$. As a result $w_{ij} = w_{ab}$ for all $i, j, a, b \in [n]$ if either $i \sim j$ and $a \sim b$ or $i \not\sim j$ and $a \not\sim b$.

With both statements proved we conclude $\Psi(\mathcal{Q}_n) \subseteq F(\mathbb{R}^3)$. Finally, as $\Psi(\mathcal{Q}_n) \subseteq F(\mathbb{R}^3)$ and $F(\mathbb{R}^3) \subseteq \Psi(\mathcal{Q}_n)$, then $\Psi(\mathcal{Q}_n) = F(\mathbb{R}^3)$ as claimed. □

The next lemma states that parameters invariant to edge adjacency preserving permutations can memorize any binary vector. In combination with Lemma 4.5 this implies any graph isomorphism class is strictly memorizable.

**Lemma 4.7.** *For $m \in [0, n]$, let $\boldsymbol{\beta} = [2, 2, 1 - 2m] \in \mathbb{R}^3$ and $\boldsymbol{\theta} = F(\boldsymbol{\beta})$. Then $E(\boldsymbol{x}^{(j)}; \boldsymbol{\theta}) - E(\boldsymbol{x}; \boldsymbol{\theta}) \geq 1$ for all $j \in [n]$ and for all $\boldsymbol{x} \in \{0, 1\}^n$ satisfying $\|\boldsymbol{x}\|_0 = m$.*

*Proof.* Let $\boldsymbol{x} \in \{0, 1\}^n$ satisfy $|supp(\boldsymbol{x})| = m \in [0, n]$. Recall from Lemma A.1 that for any $j \in [n]$

$$E(\boldsymbol{x}^{(j)}; \boldsymbol{\theta}) - E(\boldsymbol{x}; \boldsymbol{\theta}) = y_j(\boldsymbol{x}) \langle \boldsymbol{z}(\boldsymbol{x}), \boldsymbol{\theta}_j \rangle,$$

where $y_j(\boldsymbol{x}) = 1 - 2x_j$, $\boldsymbol{z}(\boldsymbol{x}) = [\boldsymbol{x}, 1]$ and $\boldsymbol{\theta}_j = [\boldsymbol{w}_j, 1]$. Furthermore,

$$\langle \boldsymbol{z}(\boldsymbol{x}), \boldsymbol{\theta}_j \rangle = \boldsymbol{x}^T \boldsymbol{w}_j + \beta_3$$

$$= \sum_{l=1}^n \mathbb{1}(l \in supp(\boldsymbol{x}) \wedge l \neq j) w_{jl} + \beta_3$$

$$= \beta_1 \sum_{l=1}^n \mathbb{1}(l \in supp(\boldsymbol{x}) \wedge l \sim j) + \beta_2 \sum_{l=1}^n \mathbb{1}(l \in supp(\boldsymbol{x}) \wedge l \not\sim j \wedge j \neq l) + \beta_3.$$

Let $c_j(\boldsymbol{x}) = \sum_{l=1}^n \mathbb{1}(l \in supp(\boldsymbol{x}) \wedge l \sim j)$ denote the number of edges of the graph adjacent to the $j$th edge. Then as

$$m = \sum_{l=1}^n \mathbb{1}(l \in supp(\boldsymbol{x}) \wedge j \neq l) + \sum_{l=1}^n \mathbb{1}(l \in supp(\boldsymbol{x}) \wedge j = l)$$

$$= \sum_{l=1}^n \mathbb{1}(l \in supp(\boldsymbol{x}) \wedge l \sim j \wedge j \neq l) + \sum_{l=1}^n \mathbb{1}(l \in supp(\boldsymbol{x}) \wedge l \not\sim j \wedge j \neq l) + \mathbb{1}(j \in supp(\boldsymbol{x}))$$

$$= \sum_{l=1}^n \mathbb{1}(l \in supp(\boldsymbol{x}) \wedge l \sim j) + \sum_{l=1}^n \mathbb{1}(l \in supp(\boldsymbol{x}) \wedge l \not\sim j \wedge j \neq l) + x_j,$$

it follows that

$$\sum_{l=1}^n \mathbb{1}(l \in supp(\boldsymbol{x}) \wedge l \not\sim j \wedge j \neq l) = m - c_j(\boldsymbol{x}) - x_j.$$

As a result, the condition that must be satisfied for all $j \in [n]$ is

$$y_j(\boldsymbol{x}) \langle \boldsymbol{z}(\boldsymbol{x}), \boldsymbol{\theta}_j \rangle = (1 - 2x_j)\left(c_j(\boldsymbol{x})(\beta_1 - \beta_2) + \beta_2(m - x_j) + \beta_3\right) \geq 1.$$

If $\beta_1 = \beta_2$ then the left-hand side simplifies to an expression which depends only on the sparsity of the representation of the graph. Under this assumption, it suffices to find a $\beta_2, \beta_3 \in \mathbb{R}$ such that

$$(1 - 2x_j)\left(\beta_2(m - x_j) + \beta_3\right) \geq 1.$$

Let $\beta_3 = 1 - \beta_2 m$, if $x_j = 0$ then

$$(1 - 2x_j)\left(\beta_2(m - x_j) + \beta_3\right) = \beta_2 m + \beta_3 = 1$$

while if $x_j = 1$ then

$$-(\beta_2(m - 1) + \beta_3) = 1 - \beta_2.$$

Therefore, with $\beta_1 = \beta_2 = 2$ and $\beta_3 = 1 - 2m$ we have

$$E(\boldsymbol{x}^{(j)}; \boldsymbol{\theta}) - E(\boldsymbol{x}; \boldsymbol{\theta}) \geq 1$$

for all $j \in [n]$. $\square$

We now make a few remarks in regard to the the construction used in the previous lemma. First, $F(2, 2, 1 - 2m)$ memorizes $\boldsymbol{x} \in \{0, 1\}^n$ iff $\|\boldsymbol{x}\|_0 = m$. Indeed, the only if aspect can be demonstrated as follows: if $\boldsymbol{x}'$ satisfies $\|\boldsymbol{x}'\| = m + \delta$ for $\delta \in \mathbb{N}_{\geq 0}$, the required inequalities become $2\delta + 1 \geq 1$ for $j \in supp(\boldsymbol{x}')$ and $-2\delta + 1 \geq 1$ for $j \notin supp(\boldsymbol{x}')$. These inequalities can only simultaneously hold if $\delta = 0$. Second

$$\|F(2, 2, 1 - 2m)\|^2 = 2v(v + 1) + (1 - 2m)^2,$$

therefore when the sparsity $m$ is proportional to $n$ then the norm scales like $\Theta(n)$.

### B.3 Constructing an invariant, small norm parameter which memorizes $k$-cliques

Our goal in this section is to show that small norm parameters exist which can memorize specific isomorphism classes. In particular, we consider the case of $k$-cliques: recall that a $k$-clique graph has a fully connected subset of $k$ vertices while the remaining $v - k$ vertices are isolated. We denote the set of representations of $k$-cliques on $v$ vertices as $\mathcal{C}_{v,k}$ and trivially note $|\mathcal{C}_{v,k}| = \binom{v}{k}$. Towards constructing low-norm invariant parameters that strictly memorize all $k$-cliques, the following lemma derives specific expressions for the energy difference derived in Lemma A.1. To state this result, for $\boldsymbol{x} \in \mathcal{C}_{v,k}$ let $\mathrm{Clique}(\boldsymbol{x}) \subset [v]$ denote the subset of the vertices of the graph which are in the fully connected subset.

**Lemma B.1.** *Let $\boldsymbol{\beta} \in \mathbb{R}^3$ and suppose $\boldsymbol{\theta} = F(\boldsymbol{\beta}) = \boldsymbol{V}\boldsymbol{\omega}$, for some $\boldsymbol{\omega} \in \mathbb{R}^q$. For $\boldsymbol{x} \in \mathcal{C}_{v,k}$ and any $j \in [n]$, define $r = |\mathrm{Clique}(\boldsymbol{x}) \cap \mathrm{Ind}^{-1}(j)| \in \{0, 1, 2\}$ as the number of vertices in the $j$th vertex pair which are also in the clique of $\boldsymbol{x}$. Then for any $j \in [n]$*

$$\boldsymbol{u}_j(\boldsymbol{x})^T \boldsymbol{\omega} = y_j(\boldsymbol{x}) z(\boldsymbol{x})^T \boldsymbol{\theta}_j = \begin{cases} \frac{\beta_2}{2}k^2 - \frac{\beta_2}{2}k + \beta_3, & r = 0, \\ \frac{\beta_2}{2}k^2 + \left(\beta_1 - \frac{3}{2}\beta_2\right)k + (\beta_2 + \beta_3 - \beta_1), & r = 1, \\ -\left(\frac{\beta_2}{2}k^2 + \left(2\beta_1 - \frac{5}{2}\beta_2\right)k + (3\beta_2 - 4\beta_1 + \beta_3)\right), & r = 2. \end{cases}$$

*Proof.* By definition

$$\boldsymbol{z}(\boldsymbol{x})^T \boldsymbol{\theta}_j = \boldsymbol{w}_j^T \boldsymbol{x} + b_j$$

$$= \sum_{l=1}^{n} w_{jl} \mathbb{1}(l \in \mathrm{supp}(\boldsymbol{x})) + \beta_3$$

$$= \beta_1 \sum_{l=1}^{n} \mathbb{1}(l \in \mathrm{supp}(\boldsymbol{x}) \wedge l \sim j) + \beta_2 \sum_{l=1}^{n} \mathbb{1}(l \in \mathrm{supp}(\boldsymbol{x}) \wedge l \nsim j \wedge l \neq j) + \beta_3$$

Observe each $j \in [n]$ can be placed in one of three distinct categories with respect to $\boldsymbol{x}$: in particular, either both, one or neither of the vertices of $j$ are in the $k$-clique of the graph represented by $\boldsymbol{x}$. Fixing an arbitrary $j \in [n]$, we denote these events in turn as $\Phi_r$ for $r \in \{0, 1, 2\}$, where

$$\Phi_r(\boldsymbol{x}) = \{j \in [n] \ : \ |\mathrm{Ind}^{-1}(j) \cap \mathrm{Clique}(\boldsymbol{x})| = r\}.$$

Note $\Phi_2(\boldsymbol{x}) = \operatorname{supp}(\boldsymbol{x})$. If $j \in \Phi_0(\boldsymbol{x})$ then neither of the vertices of $j$ are in the clique of $\boldsymbol{x}$, as a result the $j$th edge cannot be adjacent to any edge in the clique. If $j \in \Phi_1(\boldsymbol{x})$ then exactly one vertex of $j$ is in the clique, furthermore there are $k-1$ other vertices in the clique this vertex is connected to via an edge. Finally, if $j \in \Phi_2(\boldsymbol{x})$ then both of its vertices are connected via edges to $k-2$ other vertices in the clique. As a result,

$$\sum_{l=1}^{n} \mathbb{1}(l \in \operatorname{supp}(\boldsymbol{x}) \wedge l \sim j) = \begin{cases} 0, & j \in \Phi_0(\boldsymbol{x}), \\ k-1, & j \in \Phi_1(\boldsymbol{x}), \\ 2(k-2), & j \in \Phi_2(\boldsymbol{x}). \end{cases}$$

Moreover, as there are $\binom{k}{2}$ edges in total in a $k$-clique, and as if $l \in \operatorname{supp}(\boldsymbol{x})$ then $j = l$ can be true only if $j \in \operatorname{supp}(\boldsymbol{x}) \in \Phi_2(\boldsymbol{x})$, then

$$\sum_{l=1}^{n} \mathbb{1}(l \in \operatorname{supp}(\boldsymbol{x}) \wedge l \not\sim j \wedge l \neq j) = \begin{cases} \binom{k}{2}, & j \in \Phi_0(\boldsymbol{x}), \\ \binom{k}{2} - k + 1, & j \in \Phi_1(\boldsymbol{x}), \\ \binom{k}{2} - 2k + 3, & j \in \Phi_2.(\boldsymbol{x}). \end{cases}$$

As a result: if $j \in \Phi_0(\boldsymbol{x})$ then

$$\boldsymbol{z}(\boldsymbol{x})^T \boldsymbol{\theta}_j = \left( \beta_2 \frac{k(k-1)}{2} + \beta_3 \right) = \frac{\beta_2}{2} k^2 - \frac{\beta_2}{2} k + \beta_3.$$

If $j \in \Phi_1(\boldsymbol{x})$ then

$$\boldsymbol{z}(\boldsymbol{x})^T \boldsymbol{\theta}_j = \beta_1(k-1) + \beta_2 \frac{k^2 - 3k + 2}{2} + \beta_3 = \frac{\beta_2}{2} k^2 + \left( \beta_1 - \frac{3}{2} \beta_2 \right) k + (\beta_2 + \beta_3 - \beta_1).$$

Finally, if $j \in \Phi_2(\boldsymbol{x})$ then

$$\boldsymbol{z}(\boldsymbol{x})^T \boldsymbol{\theta}_j = \beta_1(2k-4) + \beta_2 \frac{k^2 - 5k + 6}{2} + \beta_3 = \frac{\beta_2}{2} k^2 + \left( 2\beta_1 - \frac{5}{2} \beta_2 \right) k + (3\beta_2 - 4\beta_1 + \beta_3).$$

To conclude, observe $y_j(\boldsymbol{x}) = (1 - 2x_{ij}) = -1$ iff $j \in \Phi_2(\boldsymbol{x})$. $\qquad \square$

We now derive a simple bound on the norm of parameters which are invariant to edge adjacency preserving permutations.

**Lemma B.2.** *Let $\boldsymbol{\beta} \in \mathbb{R}^3$ and $\boldsymbol{\theta} = F(\beta^3)$. Then*

$$\|\boldsymbol{\theta}\|^2 \leq \beta_2^2 v^4 + 2\beta_1^2 v^3 + \beta_3^2 v^2.$$

*Proof.* For any fixed edge index $r \in [n]$, note as each vertex of this edge is a member of $v - 2$ other vertex pairs then

$$\sum_{c=1}^{n} \mathbb{1}(c \sim r) = 2(v-2)$$

Moreover, as there are $\binom{v}{2}$ unordered vertex pairs in total

$$\sum_{c=1}^{n} \mathbb{1}(c \not\sim r \wedge c \neq r) = \binom{v}{2} - 2(v-2) - 1 = \frac{v^2 - 3v + 6}{2}.$$

Therefore, and also noting that $n = \binom{v}{2} \leq v^2$, we have

$$
\begin{aligned}
\|\boldsymbol{\theta}\|^2 &= \|\boldsymbol{W}\|_F^2 + \|\boldsymbol{b}\|^2 \\
&= \sum_{r=1}^n \left( \beta_1^2 \sum_{c=1}^n \mathbb{1}(c \sim r) + \beta_2^2 \sum_{c=1}^n \mathbb{1}(c \nsim r \wedge c \neq r) + \beta_3^2 \right) \\
&= n \left( \beta_1^2 2(v-2) + \beta_2^2 \left( \frac{v^2 - 3v + 6}{2} \right) + \beta_3^2 \right) \\
&\leq n(\beta_2^2 v^2 + 2\beta_1^2 v + \beta_3^2) \\
&\leq \beta_2^2 v^4 + 2\beta_1^2 v^3 + \beta_3^2 v^2.
\end{aligned}
$$

$\square$

We now present a low norm construction for memorizing $k$-cliques: in particular, Lemma 4.8 illustrates that the $k$-clique graph isomorphism class can be memorized using a parameter whose norm is $O(\sqrt{n})$. This is in contrast to the general construction used in the proof of Lemma 4.7 whose norm grew as $\Theta(n)$.

**Lemma 4.8.** *If $\boldsymbol{\beta} = [-10/k, 38/k^2, 0] \in \mathbb{R}^3$, $\boldsymbol{\theta} = F(\boldsymbol{\beta})$, and $k \geq 5$, then the following hold.*

1. *$E(\boldsymbol{x}^{(j)}; \boldsymbol{\theta}) - E(\boldsymbol{x}; \boldsymbol{\theta}) \geq 1$ for all $\boldsymbol{x} \in \mathcal{C}_{v,k}$ and all $j \in [n]$.*

2. *If $k = cv$ for some constant $c \in (0,1]$, then there exists a constant $C > 0$ such that $\|\boldsymbol{\theta}\|^2 \leq Cv$.*

*Proof.* Using Lemma B.1, and substituting $\beta_1 = -10/k$, $\beta_2 = 38/k^2$ and $\beta_3 = 0$, we obtain

$$
\boldsymbol{u}_j(\boldsymbol{x})^T \boldsymbol{\omega} = \begin{cases} 19 \left(1 - \frac{1}{k}\right), & r = 0, \\ 9 - \frac{47}{k} + \frac{38}{k^2}, & r = 1, \\ 1 + \frac{55}{k} - \frac{114}{k^2}, & r = 2. \end{cases}
$$

If $k \geq 5$, then

$$
19 \left(1 - \frac{1}{k}\right) \geq 1,
$$

while

$$
9 - \frac{47}{k} + \frac{38}{k^2} \geq 9 - \frac{47}{5} + \frac{38}{25} > 1,
$$

and

$$
1 + \frac{55}{k} - \frac{114}{k^2} \geq 1.
$$

Hence $E(\boldsymbol{x}^{(j)}; \boldsymbol{\theta}) - E(\boldsymbol{x}; \boldsymbol{\theta}) \geq 1$ for all $\boldsymbol{x} \in \mathcal{C}_{v,k}$ and all $j \in [n]$.

It remains to bound the norm. Since $F(\boldsymbol{\beta})$ assigns weight $\beta_1$ to adjacent edge pairs, weight $\beta_2$ to non-adjacent edge pairs, and has zero bias, we have

$$
\|\boldsymbol{\theta}\|^2 = \|\boldsymbol{W}\|_F^2 \leq n2(v-2)\beta_1^2 + n(n-1)\beta_2^2.
$$

Since $n = \binom{v}{2}$, $\beta_1^2 = 100/k^2$, and $\beta_2^2 = 38^2/k^4$, if $k = cv$ then

$$
n2(v-2)\beta_1^2 = O(v), \qquad n(n-1)\beta_2^2 = O(1).
$$

Therefore there exists a constant $C > 0$, depending only on $c$, such that $\|\boldsymbol{\theta}\|^2 \leq Cv$. $\square$

### B.3.1 Invariance of the full orbit HSVM solution

The following lemma states a well known result that the average orbit action of a parameter is invariant to the action of the underlying group.

**Lemma B.3.** *Let $\boldsymbol{\theta} = (\boldsymbol{W}, \boldsymbol{b}) \in \Theta$ and $\Gamma_n$ denote a subgroup of $\mathcal{P}_n$. Then $\mathrm{Proj}_{\Gamma_n}(\boldsymbol{\theta}) = \frac{1}{|\Gamma_n|} \sum_{\boldsymbol{Q} \in \Gamma_n} \boldsymbol{Q}\boldsymbol{\theta} \in \Psi(\Gamma_n)$.*

*Proof.* For typographical ease let $\boldsymbol{\theta}' = \mathrm{Proj}_{\Gamma_n}(\boldsymbol{\theta})$. Then

$$\boldsymbol{\theta}' = (\boldsymbol{W}', \boldsymbol{b}') := \frac{1}{|\Gamma_n|} \sum_{\boldsymbol{Q} \in \Gamma_n} \boldsymbol{Q}\boldsymbol{\theta} = \left( \frac{1}{|\Gamma_n|} \sum_{\boldsymbol{Q} \in \Gamma_n} \boldsymbol{Q}^T \boldsymbol{W} \boldsymbol{Q}, \frac{1}{|\Gamma_n|} \sum_{\boldsymbol{Q} \in \Gamma_n} \boldsymbol{Q}^T \boldsymbol{b} \right).$$

Given $\Gamma_n$ is a subgroup, then for any $\boldsymbol{Q}' \in \Gamma_n$

$$\begin{aligned}
\boldsymbol{Q}'\boldsymbol{\theta}' &= (\boldsymbol{Q}'^T \boldsymbol{W}' \boldsymbol{Q}', \boldsymbol{Q}'^T \boldsymbol{b}') \\
&= \left( \frac{1}{|\Gamma_n|} \sum_{\boldsymbol{Q} \in \Gamma_n} (\boldsymbol{Q}\boldsymbol{Q}')^T \boldsymbol{W}(\boldsymbol{Q}\boldsymbol{Q}'), \frac{1}{|\Gamma_n|} \sum_{\boldsymbol{Q} \in \Gamma_n} (\boldsymbol{Q}\boldsymbol{Q}')^T \boldsymbol{b} \right) \\
&= \left( \frac{1}{|\Gamma_n|} \sum_{\boldsymbol{Q} \in \Gamma_n} \boldsymbol{Q}^T \boldsymbol{W} \boldsymbol{Q}, \frac{1}{|\Gamma_n|} \sum_{\boldsymbol{Q} \in \Gamma_n} \boldsymbol{Q}^T \boldsymbol{b} \right) \\
&= \boldsymbol{\theta}',
\end{aligned}$$

therefore $\boldsymbol{\theta}' \in \Psi(\Gamma_n)$. $\qquad\square$

Using the previous lemma, we show that the full orbit HSVM solution lies in the invariant set.

**Lemma 4.9.** *Let $\boldsymbol{x}_0 \in \{0, 1\}^n$ and $\Gamma_n$ denote a subgroup of $\mathcal{P}_n$ and assume $\mathrm{Orb}(\boldsymbol{x}_0, \Gamma_n)$ can be strictly memorized. If $\boldsymbol{\theta}^* = \boldsymbol{V}\boldsymbol{\omega}^*$ where $\boldsymbol{\omega}^* = \mathrm{HSVM}_\Theta(\mathrm{Orb}(\boldsymbol{x}_0, \Gamma_n))$ then $\boldsymbol{\theta}^* \in \Psi(\Gamma_n)$.*

*Proof.* We use a symmetrization argument. To this end, with $\boldsymbol{\theta}^* = (\boldsymbol{W}^*, \boldsymbol{b}^*)$ let

$$\boldsymbol{\theta} = (\boldsymbol{W}, \boldsymbol{b}) := \frac{1}{|\Gamma_n|} \sum_{\boldsymbol{Q} \in \Gamma_n} \boldsymbol{Q}\boldsymbol{\theta}^* = \left( \frac{1}{|\Gamma_n|} \sum_{\boldsymbol{Q} \in \Gamma_n} \boldsymbol{Q}^T \boldsymbol{W}^* \boldsymbol{Q}, \frac{1}{|\Gamma_n|} \sum_{\boldsymbol{Q} \in \Gamma_n} \boldsymbol{Q}^T \boldsymbol{b}^* \right).$$

By Lemma B.3 we know that $\boldsymbol{\theta} \in \Psi(\Gamma_n)$. By the definition of $\boldsymbol{\theta}^*$ we also have

$$E(\boldsymbol{x}^{(j)}, \boldsymbol{\theta}^*) - E(\boldsymbol{x}, \boldsymbol{\theta}^*) \geq 1.$$

Therefore, using both the intertwining property (9) and Lemma A.1, for any $\boldsymbol{x} \in \mathrm{Orb}(\boldsymbol{x}_0, \Gamma_n)$ and $j \in [n]$, and with $\boldsymbol{z}(\boldsymbol{x}) = [\boldsymbol{x}, 1]$, we have

$$\begin{aligned}
E(\boldsymbol{x}^{(j)}, \boldsymbol{\theta}) - E(\boldsymbol{x}, \boldsymbol{\theta}) &= (2x_j - 1)\boldsymbol{z}(\boldsymbol{x})^T \boldsymbol{\theta}_j \\
&= \frac{1}{|\Gamma_n|} \sum_{\boldsymbol{Q} \in \Gamma_n} (2x_j - 1)\boldsymbol{z}(\boldsymbol{x})^T \boldsymbol{Q}\boldsymbol{\theta}_j^* \\
&= \frac{1}{|\Gamma_n|} \sum_{\boldsymbol{Q} \in \Gamma_n} E(\boldsymbol{x}^{(j)}, \boldsymbol{Q}\boldsymbol{\theta}^*) - E(\boldsymbol{x}, \boldsymbol{Q}\boldsymbol{\theta}^*) \\
&= \frac{1}{|\Gamma_n|} \sum_{\boldsymbol{Q} \in \Gamma_n} E(\boldsymbol{x}^{(j)}, \boldsymbol{\theta}^*) - E(\boldsymbol{x}, \boldsymbol{\theta}^*) \\
&= E(\boldsymbol{x}^{(j)}, \boldsymbol{\theta}^*) - E(\boldsymbol{x}, \boldsymbol{\theta}^*) \\
&\geq 1.
\end{aligned}$$

As a result, $\boldsymbol{\theta}$ is a feasible point of the HSVM problem (7) defined on the full orbit dataset $\mathrm{Orb}(\boldsymbol{x}_0, \Gamma_n)$. Therefore, by the definition of $\boldsymbol{\theta}^*$ it must follow that $\|\boldsymbol{\theta}^*\| \leq \|\boldsymbol{\theta}\|$. On the other hand, using the triangle inequality and the fact that $\boldsymbol{Q} \in \Gamma_n$ is a permutation, we have

$$\|\boldsymbol{\theta}\| = \|\frac{1}{|\Gamma_n|} \sum_{\boldsymbol{Q} \in \Gamma_n} \boldsymbol{Q}\boldsymbol{\theta}^*\| \leq \frac{1}{|\Gamma_n|} \sum_{\boldsymbol{Q} \in \Gamma_n} \|\boldsymbol{Q}\boldsymbol{\theta}^*\| = \|\boldsymbol{\theta}^*\|.$$

This implies $\frac{1}{2}\|\boldsymbol{\theta}^*\|^2 = \frac{1}{2}\|\boldsymbol{\theta}\|^2$, as this objective is 1-strongly convex this in turn implies $\boldsymbol{\theta}^* = \boldsymbol{\theta} \in \Psi(\Gamma_n)$. □

### B.4 Approximately invariant parameters

#### B.4.1 Approximate invariance is sufficient for generalization

The following lemma states a sufficient condition for strict memorization of an orbit dataset based on proximity to the relevant invariant space. In particular, given a graph $G \in \mathcal{G}_v$ and letting $\boldsymbol{x}_0 = \mathcal{E}_{rep}(G)$, if $\boldsymbol{\omega}^* = \mathrm{HSVM}(\mathcal{S})$, where $\mathcal{S} \subset \mathrm{Orb}(\boldsymbol{x}_0, \Phi_n)$, and $\boldsymbol{\omega}^*$ is sufficiently close to the subspace $\mathcal{Q}_n$, then $\boldsymbol{\theta}^* = \boldsymbol{E}\boldsymbol{\omega}^*$ will strictly memorize all graphs isomorphic to $G$.

**Lemma B.4.** *Let $\boldsymbol{x}_0 \in \{0,1\}^n$ satisfy $\|\boldsymbol{x}_0\| = m \in \mathbb{N}_{\geq 2}$, $\boldsymbol{\theta} = \boldsymbol{E}\boldsymbol{\omega} \in \Theta_n$ satisfy $E(\boldsymbol{x}_0^{(j)}; \boldsymbol{\theta}) - E(\boldsymbol{x}_0; \boldsymbol{\theta}) \geq 1$ for all $j \in [n]$, and $\boldsymbol{\theta}' = \boldsymbol{E}\boldsymbol{\omega}' \in \Psi(\Gamma_n)$ be such that $\|\boldsymbol{\omega} - \boldsymbol{\omega}'\| \leq \frac{1}{4\sqrt{m}}$. Then $E(\boldsymbol{x}^{(j)}; \boldsymbol{\theta}) - E(\boldsymbol{x}; \boldsymbol{\theta}) \geq \frac{1}{2}$ for all $\boldsymbol{x} \in \mathrm{Orb}(\boldsymbol{x}_0, \Gamma_n)$ and $j \in [n]$.*

*Proof.* Inspecting Lemma A.7, if $\|\boldsymbol{x}_0\| = m \in \mathbb{N}_{\geq 2}$ then $\|\boldsymbol{u}_j(\boldsymbol{x})\| \leq \sqrt{m}$ for all $\boldsymbol{x} \in \mathrm{Orb}(\boldsymbol{x}_0, \Gamma_n)$ and $j \in [n]$. By assumption $\boldsymbol{u}_j(\boldsymbol{x}_0)^T \boldsymbol{\omega} \geq 1$ for all $j \in [n]$, therefore

$$\begin{aligned}
E(\boldsymbol{x}_0^{(j)}; \boldsymbol{\theta}') - E(\boldsymbol{x}_0; \boldsymbol{\theta}') &= y_j(\boldsymbol{x}_0)\boldsymbol{z}(\boldsymbol{x}_0)^T \boldsymbol{\theta}'_j \\
&= \boldsymbol{u}_j(\boldsymbol{x}_0)^T \boldsymbol{\omega}' \\
&= \boldsymbol{u}_j(\boldsymbol{x}_0)^T \boldsymbol{\omega} - \boldsymbol{u}_j(\boldsymbol{x}_0)^T(\boldsymbol{\omega} - \boldsymbol{\omega}') \\
&\geq 1 - \|\boldsymbol{u}_j(\boldsymbol{x}_0)\|\|\boldsymbol{\omega} - \boldsymbol{\omega}'\| \\
&\geq \tfrac{3}{4}
\end{aligned}$$

for all $j \in [n]$. As $\boldsymbol{\theta}' \in \Psi(\Gamma_n)$, then Lemma 4.5 implies for any $\boldsymbol{x} \in \mathrm{Orb}(\boldsymbol{x}_0, \Gamma_n)$ that

$$E(\boldsymbol{x}^{(j)}; \boldsymbol{\theta}') - E(\boldsymbol{x}; \boldsymbol{\theta}') = \boldsymbol{u}_j(\boldsymbol{x})^T \boldsymbol{\omega}' \geq \tfrac{3}{4}.$$

Moreover, and using the same trick as before, we also observe for all $\boldsymbol{x} \in \mathrm{Orb}(\boldsymbol{x}_0, \Gamma_n)$ that

$$\begin{aligned}
E(\boldsymbol{x}^{(j)}; \boldsymbol{\theta}) - E(\boldsymbol{x}; \boldsymbol{\theta}) &= y_j(\boldsymbol{x})\boldsymbol{z}(\boldsymbol{x})^T \boldsymbol{\theta}_j \\
&= \boldsymbol{u}_j(\boldsymbol{x})^T \boldsymbol{\omega} \\
&= \boldsymbol{u}_j(\boldsymbol{x})^T \boldsymbol{\omega}' - \boldsymbol{u}_j(\boldsymbol{x})^T(\boldsymbol{\omega}' - \boldsymbol{\omega}) \\
&\geq \tfrac{3}{4} - \|\boldsymbol{u}_j(\boldsymbol{x})\|\|\boldsymbol{\omega} - \boldsymbol{\omega}'\| \\
&\geq \tfrac{1}{2}
\end{aligned}$$

for all $j \in [n]$. □

#### B.4.2 Proximity of AHSVM solution to the invariant set

**Lemma B.5.** *Let $\mathcal{S} \subset \{0,1\}^n$, $\boldsymbol{\mu}_{\mathcal{S}} = \frac{1}{|\mathcal{S}|} \sum_{\boldsymbol{x} \in \mathcal{S}} \bar{\boldsymbol{u}}(\boldsymbol{x})$ and assume $\boldsymbol{\omega}^* = \mathrm{AHSVM}(\mathcal{S})$ is feasible. Then $\boldsymbol{\omega}^* = \frac{\boldsymbol{\mu}_{\mathcal{S}}}{\|\boldsymbol{\mu}_{\mathcal{S}}\|^2}$.*

*Proof.* Note by the feasibility assumption $\frac{1}{|\mathcal{S}|} \sum_{\boldsymbol{x} \in \mathcal{S}} \bar{\boldsymbol{u}}(\boldsymbol{x}) \neq \boldsymbol{0}_q$. Forming the Lagrangian with Lagrange variable $\alpha$ we have

$$\mathcal{L}(\boldsymbol{\omega}, \alpha) = \frac{1}{2}\|\boldsymbol{\omega}\|^2 + \alpha\left(1 - \frac{1}{|\mathcal{S}|} \sum_{\boldsymbol{x} \in \mathcal{S}} \langle \bar{\boldsymbol{u}}(\boldsymbol{x}), \boldsymbol{\omega} \rangle\right).$$

Clearly this is a strongly convex objective with a unique minimizer $\boldsymbol{\omega}^*$. Zeroing the gradient with respect to $\boldsymbol{\omega}$ and rearranging gives the identity

$$\boldsymbol{\omega}^* = \alpha^* \frac{1}{|\mathcal{S}|} \sum_{\boldsymbol{x} \in \mathcal{S}} \bar{\boldsymbol{u}}(\boldsymbol{x})$$

on the solution pair $(\boldsymbol{\omega}^*, \alpha^*)$. In addition, as there is only a single constraint and the problem is feasible then the constraint must be active, meaning

$$\frac{1}{|\mathcal{S}|} \sum_{\boldsymbol{x} \in \mathcal{S}} \langle \bar{\boldsymbol{u}}(\boldsymbol{x}), \boldsymbol{\omega}^* \rangle = 1.$$

As a result

$$\|\boldsymbol{\omega}^*\|^2 = \alpha^* \frac{1}{|\mathcal{S}|} \sum_{\boldsymbol{x} \in \mathcal{S}} \langle \bar{\boldsymbol{u}}(\boldsymbol{x}), \boldsymbol{\omega}^* \rangle = \alpha^*. \tag{11}$$

Therefore

$$\frac{\boldsymbol{\omega}^*}{\|\boldsymbol{\omega}^*\|^2} = \frac{1}{|\mathcal{S}|} \sum_{\boldsymbol{x} \in \mathcal{S}} \bar{\boldsymbol{u}}(\boldsymbol{x}) = \boldsymbol{\mu}_{\mathcal{S}}.$$

As $\boldsymbol{\mu}_{\mathcal{S}} \neq \boldsymbol{0}_q$ then

$$\|\boldsymbol{\mu}_{\mathcal{S}}\|^2 = \left\| \frac{\boldsymbol{\omega}^*}{\|\boldsymbol{\omega}^*\|^2} \right\|^2 = \frac{1}{\|\boldsymbol{\omega}^*\|^2},$$

giving

$$\boldsymbol{\omega}^* = \frac{\boldsymbol{\mu}_{\mathcal{S}}}{\|\boldsymbol{\mu}_{\mathcal{S}}\|^2}$$

as claimed. $\qquad\square$

Similar to Lemma 4.9, we now show that the full orbit AHSVM solution lies on the relevant invariance space.

**Lemma B.6.** *Let $\boldsymbol{x}_0 \in \{0,1\}^n$, $\Gamma_n$ denote a subgroup of $\mathcal{P}_n$, and assume $\mathcal{O} = \mathrm{Orb}(\boldsymbol{x}_0, \Gamma_n)$ satisfies $|\mathcal{O}| = |\Gamma_n|$. Let $\boldsymbol{\omega}^* = \mathrm{AHSVM}_\Theta(\mathrm{Orb}(\boldsymbol{x}_0, \Gamma_n))$ be feasible and define $\boldsymbol{\theta}^* = \boldsymbol{V}\boldsymbol{\omega}^*$, then $\boldsymbol{\theta}^* \in \Psi(\Gamma_n)$.*

*Proof.* Again we use a symmetrization argument. To this end, with $\boldsymbol{\theta}^* = (\boldsymbol{W}^*, \boldsymbol{b}^*)$ let

$$\boldsymbol{\theta} = (\boldsymbol{W}, \boldsymbol{b}) := \frac{1}{|\Gamma_n|} \sum_{\boldsymbol{Q} \in \Gamma_n} \boldsymbol{Q}\boldsymbol{\theta}^* = \left( \frac{1}{|\Gamma_n|} \sum_{\boldsymbol{Q} \in \Gamma_n} \boldsymbol{Q}^T \boldsymbol{W}^* \boldsymbol{Q}, \frac{1}{|\Gamma_n|} \sum_{\boldsymbol{Q} \in \Gamma_n} \boldsymbol{Q}^T \boldsymbol{b}^* \right).$$

By Lemma B.3 we know that $\boldsymbol{\theta} \in \Psi(\Gamma_n)$. Let $\boldsymbol{x} \in \mathcal{O}$ and $\boldsymbol{Q} \in \Gamma_n$ be such that $\boldsymbol{Q}\boldsymbol{x} = \boldsymbol{x}_0$, and let $\pi$ denote the permutation associated with $\boldsymbol{Q}$. As $\boldsymbol{\theta} \in \Psi(\Gamma_n)$ then using the intertwining property (9)

$$\bar{\boldsymbol{u}}(\boldsymbol{x})^T \boldsymbol{\omega} = \frac{1}{n} \sum_{j=1}^n E(\boldsymbol{x}^{(j)}; \boldsymbol{\theta}) - E(\boldsymbol{x}; \boldsymbol{\theta})$$

$$= \frac{1}{n} \sum_{j=1}^n E(\boldsymbol{Q}\boldsymbol{x}^{(j)}; \boldsymbol{\theta}) - E(\boldsymbol{Q}\boldsymbol{x}; \boldsymbol{\theta})$$

$$= \frac{1}{n} \sum_{j=1}^n E((\boldsymbol{Q}\boldsymbol{x})^{(\pi(j))}; \boldsymbol{\theta}) - E(\boldsymbol{Q}\boldsymbol{x}; \boldsymbol{\theta})$$

$$= \frac{1}{n} \sum_{l=1}^n E(\boldsymbol{x}_0^{(l)}; \boldsymbol{\theta}) - E(\boldsymbol{x}_0; \boldsymbol{\theta})$$

$$= \bar{\boldsymbol{u}}(\boldsymbol{x}_0)^T \boldsymbol{\omega}.$$

As the AHSVM problem is feasible and has a single constraint then $\frac{1}{|\mathcal{O}|}\sum_{\boldsymbol{x}\in\mathcal{O}}\bar{\boldsymbol{u}}(\boldsymbol{x})^T\boldsymbol{\omega}^* = 1$. Therefore

$$
\begin{aligned}
\frac{1}{|\mathcal{O}|}\sum_{\boldsymbol{x}\in\mathcal{O}}\bar{\boldsymbol{u}}(\boldsymbol{x})^T\boldsymbol{\omega} &= \bar{\boldsymbol{u}}(\boldsymbol{x}_0)^T\boldsymbol{\omega} \\
&= \frac{1}{n}\sum_{l=1}^{n}E(\boldsymbol{x}_0^{(l)};\boldsymbol{\theta}) - E(\boldsymbol{x}_0;\boldsymbol{\theta}) \\
&= \frac{1}{n}\sum_{l=1}^{n}E\left(\boldsymbol{x}_0^{(l)};\frac{1}{|\Gamma_n|}\sum_{\boldsymbol{Q}\in\Gamma_n}\boldsymbol{Q}\boldsymbol{\theta}^*\right) - E\left(\boldsymbol{x}_0;\frac{1}{|\Gamma_n|}\sum_{\boldsymbol{Q}\in\Gamma_n}\boldsymbol{Q}\boldsymbol{\theta}^*\right) \\
&= \frac{1}{|\Gamma_n|}\sum_{\boldsymbol{Q}\in\Gamma_n}\frac{1}{n}\sum_{l=1}^{n}E(\boldsymbol{Q}\boldsymbol{x}_0^{(l)};\boldsymbol{\theta}^*) - E(\boldsymbol{Q}\boldsymbol{x}_0;\boldsymbol{\theta}^*) \\
&= \frac{1}{|\mathcal{O}|}\sum_{\boldsymbol{x}\in\mathcal{O}}\frac{1}{n}\sum_{j=1}^{n}E(\boldsymbol{x}^{(j)};\boldsymbol{\theta}) - E(\boldsymbol{x};\boldsymbol{\theta}) \\
&= \frac{1}{|\mathcal{O}|}\sum_{\boldsymbol{x}\in\mathcal{O}}\bar{\boldsymbol{u}}(\boldsymbol{x})^T\boldsymbol{\omega}^* \\
&= 1.
\end{aligned}
$$

As a result, $\boldsymbol{\theta}$ is a feasible point of the AHSVM problem defined on the full orbit dataset $\mathrm{Orb}(\boldsymbol{x}_0,\Gamma_n)$. Therefore, by the definition of $\boldsymbol{\theta}^*$ it must follow that $\|\boldsymbol{\theta}^*\| \leq \|\boldsymbol{\theta}\|$. On the other hand, using the triangle inequality and the fact that $\boldsymbol{Q}\in\Gamma_n$ is a permutation, we have

$$
\|\boldsymbol{\theta}\| = \|\frac{1}{|\Gamma_n|}\sum_{\boldsymbol{Q}\in\Gamma_n}\boldsymbol{Q}\boldsymbol{\theta}^*\| \leq \frac{1}{|\Gamma_n|}\sum_{\boldsymbol{Q}\in\Gamma_n}\|\boldsymbol{Q}\boldsymbol{\theta}^*\| = \|\boldsymbol{\theta}^*\|.
$$

This implies $\frac{1}{2}\|\boldsymbol{\theta}^*\|^2 = \frac{1}{2}\|\boldsymbol{\theta}\|^2$, as this objective is 1-strongly convex this in turn implies $\boldsymbol{\theta}^* = \boldsymbol{\theta} \in \Psi(\Gamma_n)$. □

The lemma below bounds the difference between the sample AHSVM solution and the population AHSVM solution, leveraging only the boundedness of the data.

**Lemma 4.10.** *Let $\mathcal{O} \subseteq \{0,1\}^n$ satisfy $\|\boldsymbol{x}\|_0 \leq m \in \mathbb{N}_{\geq 2}$ for all $\boldsymbol{x} \in \mathcal{O}$ and assume $\boldsymbol{\omega}^* = \mathrm{HSVM}(\mathcal{O})$ is feasible. Consider a random sample $\mathcal{S} = (\boldsymbol{x}_i)_{i=1}^N$ where $\boldsymbol{x}_i \sim U(\mathcal{O})$ are mutually i.i.d. and define $\boldsymbol{\omega}_\mathcal{O} = \mathrm{AHSVM}(\mathcal{O})$ and $\boldsymbol{\omega}_\mathcal{S} = \mathrm{AHSVM}(\mathcal{S})$. For $\delta \in (0,1]$ and $\epsilon \in \mathbb{R}_{>0}$, if $N \gtrsim \epsilon^{-2}\|\boldsymbol{\omega}^*\|^4 m \log(1/\delta)$ then $\|\boldsymbol{\omega}_\mathcal{S} - \boldsymbol{\omega}_\mathcal{O}\| \leq \epsilon$ with probability at least $1 - \delta$.*

*Proof.* Let $\boldsymbol{\mu} = \mathbb{E}[\bar{\boldsymbol{u}}(\boldsymbol{x})]$ where $\boldsymbol{x} \sim U(\mathcal{O})$, then $\boldsymbol{\mu} = \frac{1}{|\mathcal{O}|}\sum_{\boldsymbol{x}\in\mathcal{O}}\bar{\boldsymbol{u}}(\boldsymbol{x})$. In addition, let $\hat{\boldsymbol{\mu}}_\mathcal{S} = \frac{1}{|\mathcal{S}|}\sum_{\boldsymbol{x}\in\mathcal{S}}\bar{\boldsymbol{u}}(\boldsymbol{x})$. Then using Lemma B.5 we have

$$
\boldsymbol{\omega}_\mathcal{O} = \frac{\boldsymbol{\mu}}{\|\boldsymbol{\mu}\|^2}, \quad \boldsymbol{\omega}_\mathcal{S} = \frac{\hat{\boldsymbol{\mu}}_\mathcal{S}}{\|\hat{\boldsymbol{\mu}}_\mathcal{S}\|^2}.
$$

Taking norms this clearly also implies

$$
\|\boldsymbol{\mu}\| = \frac{1}{\|\boldsymbol{\omega}_\mathcal{O}\|}, \quad \|\hat{\boldsymbol{\mu}}_\mathcal{S}\| = \frac{1}{\|\boldsymbol{\omega}_\mathcal{S}\|},
$$

Observe by definition that $\boldsymbol{\omega}^*$ satisfies $E(\boldsymbol{x}^{(j)};\boldsymbol{\theta}^*) - E(\boldsymbol{x};\boldsymbol{\theta}^*) = \langle\boldsymbol{u}_j(\boldsymbol{x}),\boldsymbol{\omega}^*\rangle \geq 1$, therefore for any $S \subseteq \mathcal{O}$ we have

$$
\frac{1}{|S|}\sum_{\boldsymbol{x}\in S}\langle\bar{\boldsymbol{u}}(\boldsymbol{x}),\boldsymbol{\omega}^*\rangle = \frac{1}{n|S|}\sum_{\boldsymbol{x}\in S}\sum_{j=1}^{n}\langle\boldsymbol{u}_j(\boldsymbol{x}),\boldsymbol{\omega}^*\rangle \geq 1.
$$

As a result we have both $\boldsymbol{\omega}^* \in \mathcal{F}_A(\mathcal{S})$ and $\boldsymbol{\omega}^* \in \mathcal{F}_A(\mathcal{O})$, which in turn implies $\|\boldsymbol{\omega}^*\| \geq \|\boldsymbol{\omega}_\mathcal{S}\|$ and $\|\boldsymbol{\omega}^*\| \geq \|\boldsymbol{\omega}_\mathcal{O}\|$. Defining $f(\boldsymbol{x}) = \boldsymbol{x}/\|\boldsymbol{x}\|^2$ for any $\boldsymbol{x} \in \mathbb{R}^q$, then applying Lemma A.8 this gives

$$
\|\boldsymbol{\omega}_\mathcal{S} - \boldsymbol{\omega}_\mathcal{O}\| = \|f(\hat{\boldsymbol{\mu}}_\mathcal{S}) - f(\boldsymbol{\mu})\| \leq \frac{\|\hat{\boldsymbol{\mu}}_\mathcal{S} - \boldsymbol{\mu}\|}{\min\{\|\boldsymbol{\omega}_\mathcal{O}\|^{-2},\|\boldsymbol{\omega}_\mathcal{S}\|^{-2}\}} \leq 3\|\boldsymbol{\omega}^*\|^2\|\hat{\boldsymbol{\mu}}_\mathcal{S} - \boldsymbol{\mu}\|.
$$

Observe

$$\|\hat{\boldsymbol{\mu}}_{\mathcal{S}} - \boldsymbol{\mu}\| = \|\frac{1}{N} \sum_{i=1}^{N} (\bar{\boldsymbol{u}}(\boldsymbol{x}) - \boldsymbol{\mu})\|,$$

clearly $\bar{\boldsymbol{u}}(\boldsymbol{x}) - \mu$ is a centered random vector, moreover for any $\boldsymbol{x} \in \{0, 1\}$

$$\|\bar{\boldsymbol{u}}(\boldsymbol{x}) - \mu\| \leq \|\bar{\boldsymbol{u}}(\boldsymbol{x})\| + \|\boldsymbol{\mu}\|$$
$$= \|\bar{\boldsymbol{u}}(\boldsymbol{x})\| + \left\|\frac{1}{|\mathcal{O}|} \sum_{\boldsymbol{x}' \in \mathcal{O}} \bar{\boldsymbol{u}}(\boldsymbol{x}')\right\|$$
$$\leq \|\bar{\boldsymbol{u}}(\boldsymbol{x})\| + \frac{1}{|\mathcal{O}|} \sum_{\boldsymbol{x}' \in \mathcal{O}} \|\bar{\boldsymbol{u}}(\boldsymbol{x}')\|$$
$$\leq 2\sqrt{m},$$

where the last inequality follows from Lemma A.7. We now deploy Lemma A.9, a specialization of (Pinelis, 1994, Th, 3.5). In particular, given some $\epsilon \in \mathbb{R}_{\geq 0}$ and letting $S_N = \sum_{i=1}^{N} (\bar{\boldsymbol{u}}(\boldsymbol{x}) - \boldsymbol{\mu})$ then

$$\mathbb{P}\left(\|\hat{\boldsymbol{\mu}}_{\mathcal{S}} - \boldsymbol{\mu}\| \geq \epsilon\right) = \mathbb{P}(\|S_N\| \geq N\epsilon) \leq \exp\left(-\frac{N\epsilon^2}{4m}\right)$$

As a result, for $\delta \in (0, 1]$, if $N \geq \frac{m}{4\epsilon^2} \log(1/\delta)$ then

$$\|\boldsymbol{\omega}_{\mathcal{S}} - \boldsymbol{\omega}_{\mathcal{O}}\| \leq 3\|\boldsymbol{\omega}^*\|^2 \epsilon$$

with probability at least $1 - \delta$. To arrive at the claimed result we substitute $\epsilon$ for $\frac{\epsilon}{3\|\boldsymbol{\omega}^*\|^2}$. □

## C    Additional Experiments and Further Preliminary Results

### C.1    Further training and test accuracy curves

Fig. 4 shows test accuracy versus training sample size, with mean and min–max over 10 trials, for Hopfield networks trained by MEF, Perceptron, and Delta (the latter two used only as baselines; see Appendix A.2). For small graphs ($v = 8$) we enumerate the full isomorphism class and report the true accuracy, i.e., the fraction of the class memorized. For larger graphs ($v = 20$), accuracy is estimated on an independent random sample of 1000 graphs. Note, within our hyperparameter range, the Delta rule using Adam failed on the $k$-clique class, whereas MEF learned all classes and was insensitive to optimizer choice.

### C.2    Further parameter heatmaps

Fig. 5 shows the weight matrices for MEF and Delta on clique and Paley graph data. We observe that the Delta rule also returns solutions which approach the invariant space as the sample size $N$ increases.

### C.3    Sample Complexity for Robust Exponential Memory

Robust exponential memory in Hopfield networks can be equivalently viewed through the lenses of error-correcting codes and the recovery of latent combinatorial structure under noise. Prior work has shown, specifically for $k$-cliques, that Hopfield networks can store entire isomorphism classes of patterns with large basins of attraction, achieving asymptotically optimal noise tolerance. As discussed in Section 4.3, Theorem 3.2, Lemma 4.7, and Lemma 4.8 together imply that a polynomial number of randomly sampled representatives from an isomorphism class suffices for MEF-trained Hopfield networks to memorize nearly all elements of that class. This provides theoretical evidence that MEF learning can recover invariant structure from sparse data, supporting and extending earlier conjectures on robust generalization in structured memory models. An important direction for future work is to determine whether polynomial sample complexity suffices to guarantee memorization of all elements of an isomorphism class, and whether such learned networks

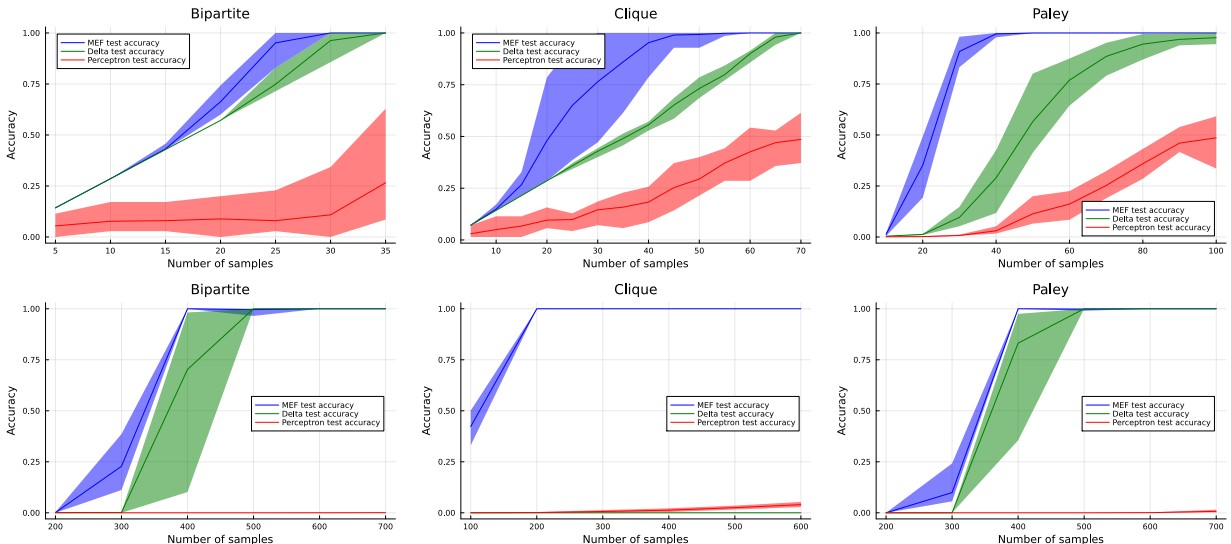

Figure 4: **Test accuracy vs. training sample size for isomorphism classes at two scales.** Top row: $v = 8$ (isomorphism class sizes: bipartite 35, Paley 2520). Bottom row: $v = 20$ (for reference class size for bipartite is 92,378). Curves show mean and min-max over 10 trials. Networks are trained with Perceptron, Delta (MSE), and MEF learning rules.

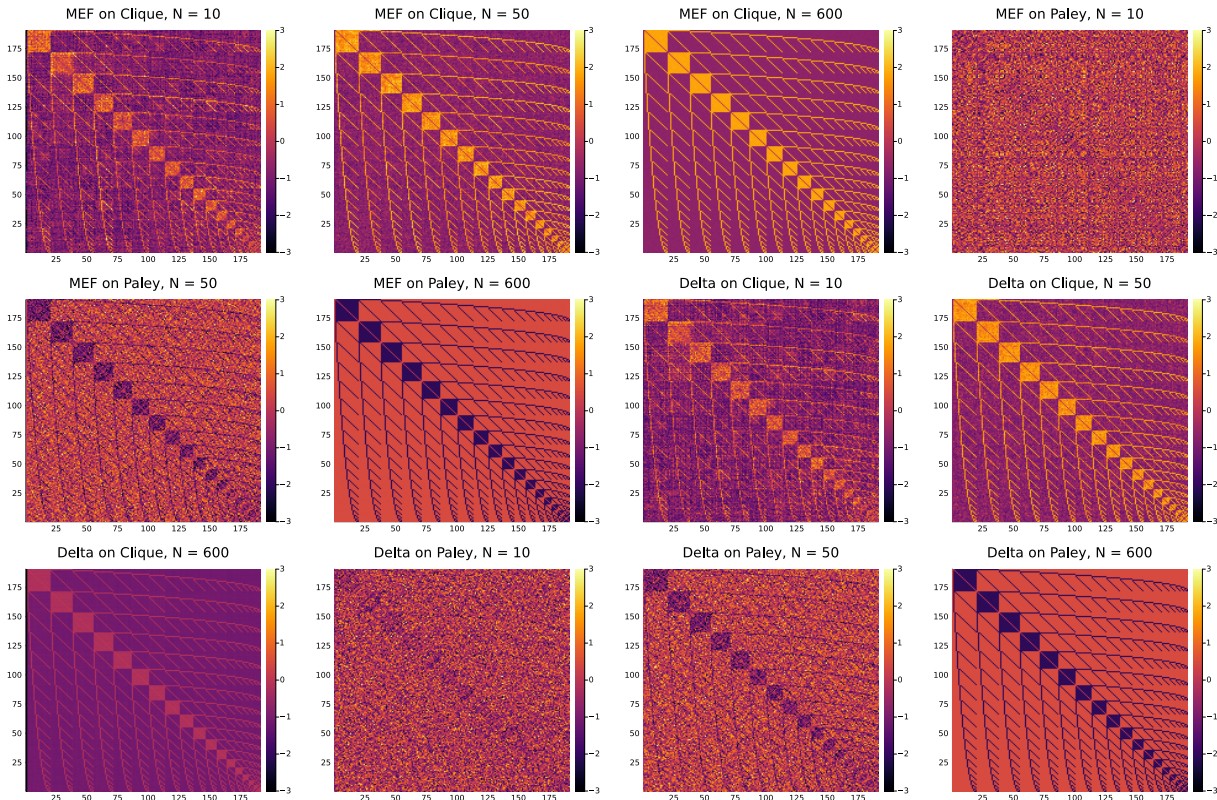

Figure 5: **Weights found by MEF and Delta on clique and Paley graph data while varying $N$:** networks where trained on samples from isomorphism class of 10-cliques and Paley graphs on $v = 20$ vertices with sample size ranging from $N = 10$ to 600.

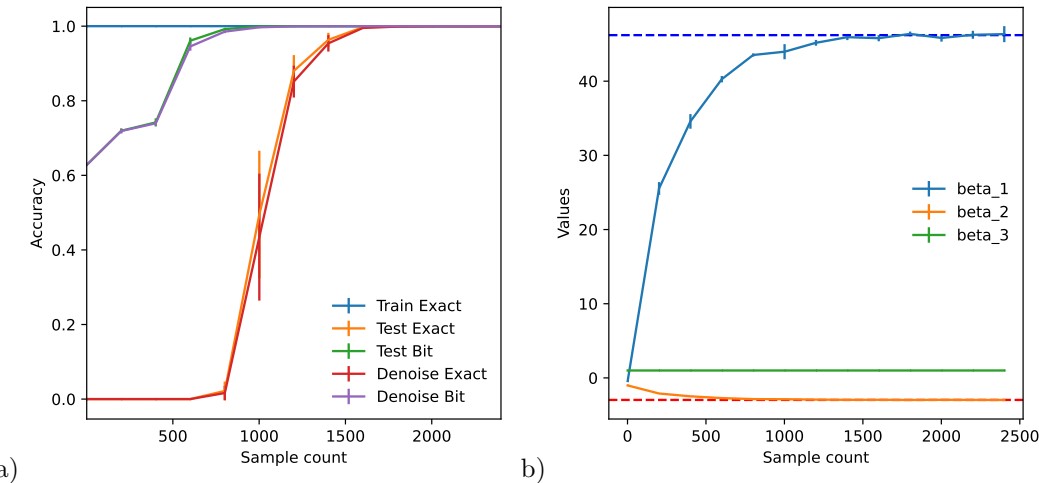

Figure 6: **Generalization on the Hidden Clique Problem.** a) Generalization and denoising accuracy (exact / average bits) are plotted as a function of number of 50-clique samples (in 100-vertex graphs; $n = 4950$ bit networks) for MEF-trained HNs. Accuracy for generalization was computed using 10000 novel graphs as Test set. Denoising accuracy was computed by corrupting 5% bits in these 10000 and evaluating correctness of the attractor when dynamics is initialized at the noisy patterns (5 trials, standard deviation error bars). b) For each type of learned parameter (weights indexed by adjacent/non-adjacent edges, or thresholds), we plot their average over number of training samples (normalized so that thresholds have mean absolute value 1). Dotted horizontal lines are parameter type averages from training on the largest number of samples.

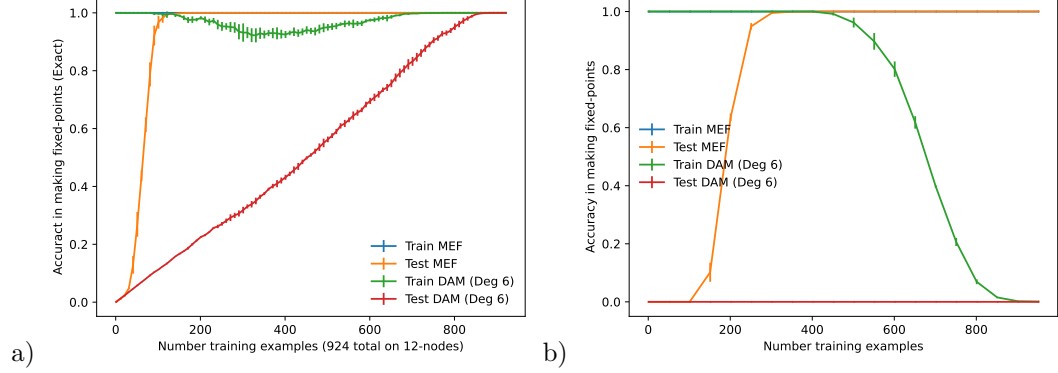

Figure 7: **DAM models trained on cliques.** We compare the generalization performance between DAMs and MEF-trained HNs for a) the 6-clique problem on graphs with $v = 12$ vertices and b) the 16-clique problem on graphs with $v = 32$ vertices (496-bit networks). The Train accuracy is the percentage of clique training samples that are neural network attractors. For a, the Test accuracy is the percentage of all 6-cliques that are fixed-points, and for b, it is that for a random set of 10000 16-cliques. Scores are averaged over four trials, with standard deviation error bars.

can reliably perform structured recovery tasks associated with that class under noise. The clique family and the Hidden Clique Problem serve as a concrete illustration of this phenomenon, but the underlying mechanism applies more broadly to invariant families of combinatorial patterns. For an example of this robust generalization, see Fig. 6, which plots over sample count both the generalization and denoising accuracy of very large HNs trained with MEF ($v = 100$; $n = 4950$ bits).

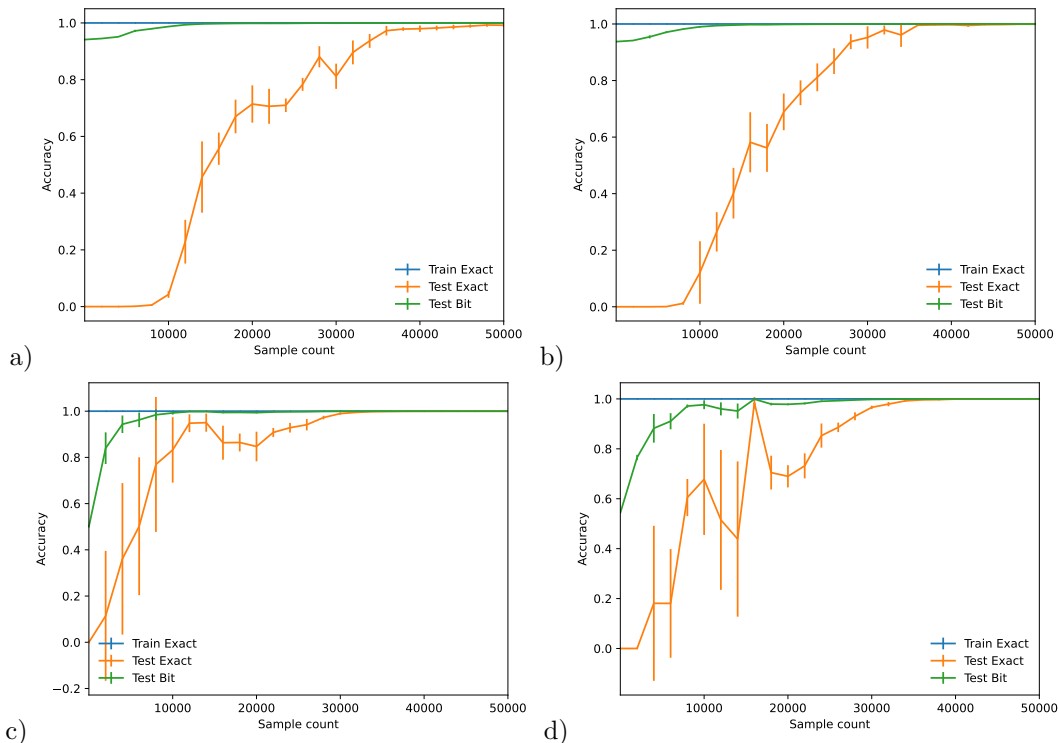

Figure 8: **Generalization for other graph classes.** Accuracy of MEF-trained HN networks generalizing as a function of number of training samples for: a) Chain graph with $v = 32$, $k = 16$ (496-bit), b) Cycle graph with $v = 32$, $k = 16$ (496-bit), c) Bipartite graph with $v = 32$, $k = 16$ (496-bit), and d) Johnson graph $J(7, 3)$ on $v = 35$ vertices (595-bit). Averaged over 10 trials, with standard deviation error bars. Test sets consisted of 10000 novel graphs chosen randomly from the isomorphism class.

## C.4 Clique generalization in DAMs

We conducted experiments comparing the generalization performance of DAMs with that of MEF when presented with increasing numbers of cliques as training data. We used the architecture and algorithm described in Krotov & Hopfield (2016) for polynomial degree 6 in the energy function. The results in Fig. 7 show very different generalization behaviors. These preliminary investigations suggest that DAMs do not generalize well in the setting of cliques, but much more work is needed to understand their behavior.

## C.5 Graph Families

We also see generalization occurring for several other families of graphs. Fig. 8 shows Train and test generalization (bit and exact) accuracy for generalization in the four graph classes: chain, cycle, bipartite, and Johnson (see also Fig. 3). In these cases the integer $k$ corresponds to number of vertices in the corresponding chain, cycle, or bipartite subset. Although most graph classes appear to have monotonic increases in accuracy as a function of number of training samples, it is interesting that for some settings there is a "double ascent/descent" property, which can be seen in Fig. 8d for the Johnson graph family. Another example of this phenomenon for the case of cliques appears in Section C.6. We also list several experiments with circulant graphs in Fig. 9.

## C.6 Double Descent Phenomenon

In our experiments we observed the interesting phenomenon that sometimes the Test error decreased with number of training samples, but then increased for a time, only to decrease again to perfect accuracy (zero

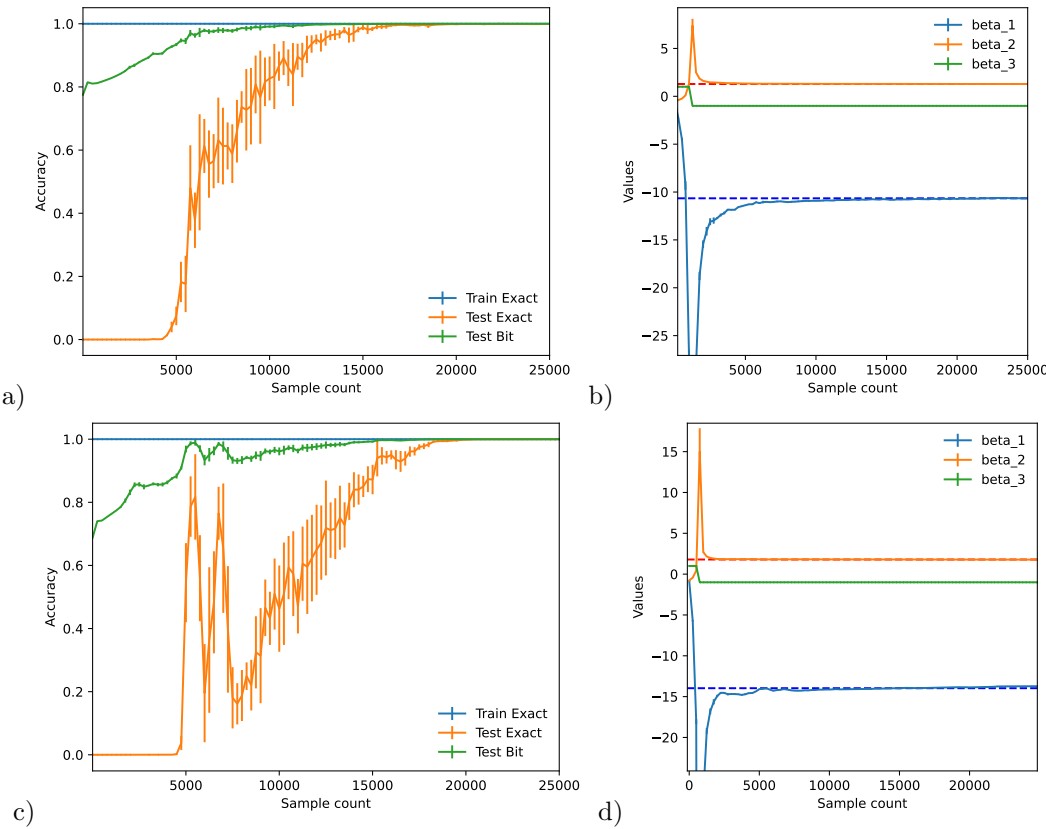

Figure 9: **Generalization for Circulant graphs.** Accuracy of MEF-trained HN networks generalizing as a function of number of training samples for: a) Circulant graph with $v = 32$, jump sequence $[2, 4]$, b) For each type of learned parameter (weights indexed by adjacent/non-adjacent edges, or thresholds) in data from a, we plot their average over number of training samples (normalized so that thresholds have mean absolute value 1; dotted horizontal lines are parameter type averages from training on the largest number of samples), c) Circulant graph with $v = 32$, jump sequence $[2, 3, 5]$, and d) Correspondingly as in b for graph in c. Plots are averaged over 10 trials, with standard deviation error bars. Test sets consisted of 10000 novel graphs chosen randomly from the isomorphism class.

error). We give two examples in Fig. 10 for bit error between attractors and samples for the case of cliques. For cliques, we note that this phenomenon seems to occur with smaller $k$ relative to $v$. It is also interesting to see that when $k$ is small, it takes many more samples to learn the full isomorphism class. In other words, it seems that the easiest $k$-clique isomorphism learning problem is when $k$ is near $v/2$.

More examples of multiple (double and triple) ascent in accuracy can be found in Fig. 8cd and Fig. 9c for exact error (percentage of samples that are fixed points of the dynamics) in the case of Johnson and Circulant graphs. We postpone theoretical analysis of these findings for future work.

### C.7 The Hopfield Network Not Graph Isomorphic Check (HNNGIC)

Lemma 4.7 implies any isomorphism class of a graph can be stored in a Hopfield network. This prompts investigation into the potential for using Hopfield networks to check for graph isomorphisms, a fundamental and important problem in computer science. To this end we propose Algorithm 1, which we refer to as the *Hopfield Network Not Graph Isomorphic Check* (HNNGIC). As the name suggests, this algorithm provides a check if two graphs are not graph isomorphic, returning true in certain cases when they are not graph isomorphic and unknown, otherwise.

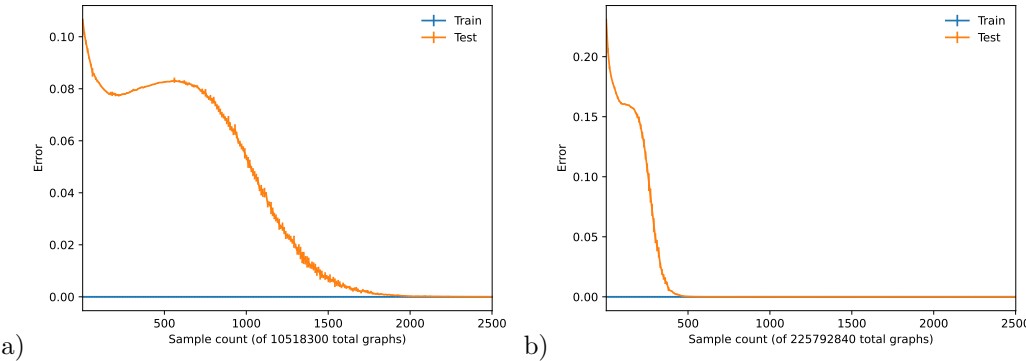

a)  b)

Figure 10: **Double Descent Phenomenon.** a) We trained Hopfield networks using MEF with increasing numbers of training samples for the case of learning cliques with $v = 32, k = 8$ and find that average test bit error (on 10000 random cliques) decreases with number of training samples, then mysteriously increases, only to decrease again to zero. (5 trials for choices of randomly samples train clique data, standard deviation error bars). b) Same as in a, but for graphs with $v = 32, k = 12$. Note that the double descent phenomenon is less apparent for this larger choice of $k$.

---

**Algorithm 1:** Hopfield Network Not Graph Isomorphic Check (HNNGIC)

---

**Input:** two graphs $\boldsymbol{x}_1, \boldsymbol{x}_2 \in \{0,1\}^n$ and computational budget $B$
**Output:** True or Unknown

**Step 1:** minimize $L(F(\boldsymbol{\beta}); \boldsymbol{x}_1)$ within computational budget $B$, return $\boldsymbol{\beta}^* \in \mathbb{R}^3$;
**Step 2: if** $H(\boldsymbol{x}_1; F(\boldsymbol{\beta}^*)) = \boldsymbol{x}_1$ **then**
    **if** $H(\boldsymbol{x}_2; F(\boldsymbol{\beta}^*)) \neq \boldsymbol{x}_2$ **then**
        **return** True
    **end**
**else**
    **return** Unknown
**end**

---

The idea behind this algorithm is simple: given two graphs, we pick one, i.e., $\boldsymbol{x}_1$, arbitrarily at random. We then attempt to train the Hopfield network by minimizing the energy flow defined on this single graph, but restrict the parameters to lie on the edge adjacency invariant subspace $\Psi(\mathcal{Q}_n)$. If the resulting invariant parameters $F(\boldsymbol{\beta}^*)$ strictly memorize $\boldsymbol{x}_1$ then this implies every point in the orbit of $\boldsymbol{x}_1$ under graph isomorphism is also strictly memorized. Therefore, if $\boldsymbol{x}_2$ is graph isomorphic to $\boldsymbol{x}_1$, then it must be a fixed point. As a result, $\boldsymbol{x}_2$ and $\boldsymbol{x}_1$ cannot be graph isomorphic if $\boldsymbol{x}_2$ is not a fixed point of an invariant Hopfield network which stores $\boldsymbol{x}_1$. Note, that if $\boldsymbol{x}_2$ is a fixed point, it does not follow that $\boldsymbol{x}_1$ and $\boldsymbol{x}_2$ are isomorphic. Indeed, there may be other fixed points, i.e., "spurious states", in the landscape, not related to the orbit of $\boldsymbol{x}_1$. Also note that Lemmas 4.7 and 4.5 imply that there is always a 3-parameter network storing any graph; in particular, given enough computation, we are guaranteed to find an approximation of $\boldsymbol{\beta}^*$ that is sufficient to store $x_1$.

