# OpenReview forum: "Implicit Bias and Invariance: How Hopfield Networks Efficiently Learn Graph Orbits"
_TMLR — Under review for TMLR_

### Review · Reviewer_4ohf · 2026-07-01

**Summary Of Contributions:**

This paper studies how invariance can emerge implicitly in classical Hopfield networks trained on group-structured graph data. The main contribution is to show that memorization in Hopfield networks can be reformulated as a linear feasibility / hard-margin SVM-type problem, and that minimizing the MEF objective with gradient descent has an implicit bias toward minimum-norm, max-margin-like solutions. The paper then connects this implicit norm bias to orbit generalization: when the data consist of graph isomorphism classes, low-norm solutions tend to approach an invariant parameter subspace, allowing the network to generalize from a small random subset of an orbit to unseen isomorphic graphs.

A particularly interesting technical contribution is the characterization of a three-dimensional invariant parameter subspace for graph edge representations, where weights only depend on whether two edge-coordinates are identical, adjacent, or non-adjacent, plus a shared bias term. The paper further provides explicit constructions showing that graph isomorphism classes can be memorized by Hopfield networks, along with sample-complexity guarantees and empirical results on cliques, bipartite graphs, Paley graphs, and Johnson graphs.

The main strengths are the clean connection between Hopfield memorization, max-margin learning, implicit bias, and emergent invariance; the paper also gives a conceptually appealing explanation of few-shot-to-orbit generalization. The main weaknesses are that the experiments are mostly synthetic graph-orbit settings, so the practical scope is not yet clear.

**Additional Comments:**

I am not an expert in classical Hopfield networks, so my evaluation of the Hopfield-specific technical details is based mainly on the clarity and consistency of the paper’s arguments rather than deep prior expertise in this model class.

**Audience:**

Yes

**Audience Explanation:**

The paper should be of interest to researchers working on implicit bias, invariance/equivariance, associative memory, graph symmetries, and the theory of generalization. The setting is intentionally minimal, but this is also one of the paper’s strengths: classical Hopfield networks provide a clean mathematical testbed for studying how symmetry can emerge from optimization and data structure rather than being built into the architecture.

The result is also conceptually relevant beyond Hopfield networks. The paper suggests that minimum-norm or max-margin implicit bias can act as a kind of hidden symmetry regularizer when training data are sampled from a group orbit. This perspective could be useful for researchers studying why non-equivariant models sometimes learn approximately invariant representations from structured data.

The audience may be narrower than for a practical graph-learning paper, since the experiments are synthetic and the model is classical rather than state-of-the-art. However, for the TMLR audience interested in theoretical mechanisms of representation learning and generalization, the findings are likely to be relevant.

**Broader Impact Concerns:**

I do not see major broader-impact concerns. The work is theoretical and uses synthetic graph data, with no human subjects or sensitive data. The existing broader impact statement is sufficient.

**Claims And Evidence:**

Yes

**Claims Explanation:**

Overall, the main claims are supported by a combination of formal derivations, theoretical results, and experiments. The paper clearly derives the equivalence between strict Hopfield memorization and a system of linear inequalities, which justifies the connection to hard-margin SVM solutions. The implicit-bias claim for MEF-GD is also reasonably supported by rewriting the MEF objective as an exponential loss over induced linear features and invoking known max-margin convergence results.

The graph-specific claims are supported by the characterization of the invariant parameter subspace and by explicit constructions showing that graph isomorphism classes can be memorized. The experiments provide consistent evidence that MEF generalizes better than Perceptron and Delta-style learning rules in the tested graph-orbit settings, and the observed learned weights visually align with the predicted invariant structure.

**Requested Changes:**

The paper should emphasize that the contribution is mainly theoretical and conceptual, not a practical replacement for equivariant graph architectures or modern Hopfield networks.

---

> ### Author Response · Authors · 2026-07-06
>
> We thank the reviewer for their careful and positive assessment. We are glad that the reviewer found the connection between Hopfield memorization, max-margin learning, implicit bias, and emergent invariance clear and conceptually appealing. We agree with the requested change and will adjust the introduction/abstract accordingly to better emphasize that the contribution is conceptual in nature, and does not offer a practical alternative to equivariant graph architectures or modern Hopfield networks.

---

### Review · Reviewer_rygX · 2026-07-02

**Summary Of Contributions:**

This paper studies how group invariance can emerge implicitly in classical Hopfield networks trained on graph isomorphism classes. It connects minimum energy flow training to minimum-norm hard-margin solutions, derives a polynomial sample-complexity bound for generalizing to unseen orbit members, and characterizes a three-dimensional invariant parameter subspace. Experiments on several graph families support the observed few-shot orbit generalization and convergence toward invariant parameters.

**Audience:**

Yes

**Audience Explanation:**

Yes. The paper should interest researchers working on implicit bias, symmetry and equivariance, associative memory, graph learning, and statistical learning theory.

**Claims And Evidence:**

Yes

**Claims Explanation:**

The main mathematical claims are generally well supported, particularly the MEF–HSVM connection, the invariant parameter construction, and the sample-complexity result. The experiments also consistently show improved generalization to unseen orbit members.

**Requested Changes:**

My comments are as follows:
- It would be helpful to clarify the distinction between generalizing to unseen members of an orbit and identifying an isomorphism class more broadly.
- The empirical evaluation could be strengthened by including non-isomorphic graphs, particularly graphs with the same numbers of vertices and edges, to assess the prevalence of spurious memories.
- A direct quantitative measure of the distance between learned parameters and the invariant subspace would make the evidence for emergent invariance more convincing.
- Additional details on the experimental optimizers and their connection to the analyzed gradient-descent dynamics would improve the alignment between theory and experiments.

---

> ### Author Response · Authors · 2026-07-06
>
> We thank the reviewer for the careful and constructive comments. We are glad that the reviewer found the MEF–HSVM connection, invariant-parameter construction, sample-complexity result, and empirical evidence convincing. We address the requested changes below.
>
> - *Clarify the distinction between generalizing to unseen members of an orbit and identifying an isomorphism class more broadly.* We would be happy to clarify this. To avoid misunderstanding, our current interpretation is that the reviewer is distinguishing orbit generalization from the stronger task of selective class identification. By orbit generalization, we mean that a network trained on samples from a fixed isomorphism class memorizes unseen members of that same orbit, either with high probability under a distribution on the orbit or, in stronger cases, across the full orbit. By selective class identification, we mean the stronger “all and only” task of memorizing the target isomorphism class while rejecting non-isomorphic graphs, including graphs with the same numbers of vertices and edges. Our results are intended to address orbit generalization, not this stronger selective-identification task. We will make this distinction explicit in the paper. If the reviewer intended a different distinction, we would be grateful for clarification.
>
> - *Include non-isomorphic graphs, particularly graphs with the same numbers of vertices and edges, to assess spurious memories.* We agree that this is a useful diagnostic, especially because orbit generalization does not by itself imply selective memorization of only the target isomorphism class. We already have preliminary experiments in this direction, and will include or summarize them where space permits. More broadly, we will revise the discussion to make clear that our main guarantees concern memorization/generalization on the target orbit, while quantifying the full landscape of spurious memories is an important but separate question.
>
> - *A direct quantitative measure of the distance between learned parameters and the invariant subspace would make the evidence for emergent invariance more convincing.* We agree that such a measure would make the evidence clearer. The current histograms and heatmaps were intended as qualitative evidence that learned parameters concentrate near the invariant subspace, but a normalized projection residual, such as distance to the three-dimensional invariant subspace after orthogonal projection, would be a natural quantitative diagnostic. We will consider adding this diagnostic from the existing experimental runs where feasible, and in any case will clarify that the present evidence is qualitative.
>
> - *Additional details on the experimental optimizers and their connection to the analyzed gradient-descent dynamics would improve the alignment between theory and experiments.* We agree that the optimizer details should be easier to locate and will expand the experimental details appendix accordingly. In particular, we will think how to more clearly state the optimizer choices and optimizer parameter configuration. We remark that the anonymous code repository also contains the full implementation details for our experiments.

---

### Review · Reviewer_C1hJ · 2026-07-02

**Summary Of Contributions:**

The paper studies the emergence of invariance in classical Hopfield networks (HNs) trained on graph isomorphism classes.
They state three main contributions. 1) the rank-3 subspace of parameters invariant to edge-adjacency-preserving permutations, and show this subspace suffices to memorize any graph isomorphism class (extending Hillar & Tran 2018, which was restricted to cliques). 2) they show that gradient descent on the MEF (minimum energy flow) loss converges in direction to the max-margin solution, building on Soudry et al. 2018, and derive a polynomial sample complexity bound for generalizing to the full orbit. 3) they show empirically, and partially in theory that learned parameters converge toward this invariant subspace as sample size grows.

Main strength: the link between implicit bias and emergent invariance is clean, and the structural result (rank-3 subspace) is a solid contribution in its own right.

Main weakness: the motivation and community interest are never clearly stated. The system under study (classical HNs) is a toy model and not linked to the modern architectures, and the paper does not explicitly say who this result is for or why it matters beyond its own construction.

**Audience:**

Yes

**Audience Explanation:**

I believe this could be of interest for researchers working on implicit bias as a theoretical mechanism, for whom this minimal, fully analyzable system has value in its own right.

That said, I state this with caution: I felt that the paper never makes the effort to clearly explain this interest. The abstract uses fairly general language ("a unifying mechanism for generalization in Hopfield networks") that suggests a broader scope than what the content actually demonstrates (everything is specific to graph orbits under classical HNs). The introduction does not orient the reader on why this mechanism is worth studying on this particular system, or what is learned that would not already be expected from combining Hillar et al. 2021 and Soudry et al. 2018. The connection to broader questions (modern architectures, practical generalization) remains implicit and confined to a few sentences in the conclusion. Without this positioning work, I remain uncertain about who, concretely, this paper is meant for and why, hence my request for clarification below.

**Broader Impact Concerns:**

No concerns.

**Claims And Evidence:**

Yes

**Claims Explanation:**

I should note upfront that this topic is quite far from my area of expertise, so my assessment of the proofs remains cautious.
I did my best to check the central proofs and found no errors. The experiments on synthetic data support the theoretical results.

**Requested Changes:**

Explicitly state, in 1-2 paragraphs in the introduction, the motivation and intended audience: why study implicit bias specifically in classical HNs rather than in a more general or more practically relevant setting? What does this work add beyond the direct combination of Hillar et al. 2021 and Soudry et al. 2018?

Recalibrate the abstract and conclusion so as not to suggest a broader scope ("unifying mechanism for generalization") than what is demonstrated (specific to graph orbits under classical HNs).

---

> ### Author Response · Authors · 2026-07-06
>
> We thank the reviewer for their feedback and suggestions. We are glad that the link between implicit bias and emergent invariance was clear, and that the reviewer views the rank-3 subspace as a solid contribution. The main concerns appear to be the motivation, intended audience, relation to prior work, and scope of the claims. We address these in turn.
>
> - *The motivation is never clearly stated*. For many graph families, the isomorphism class is exponentially large in the number of vertices. Nevertheless, despite having no explicit graph-isomorphism invariance in the architecture, standard Hopfield training procedures can empirically generalize from a vanishingly small fraction of this class, and our results provide polynomial sample-complexity guarantees in this setting. We view this as a nontrivial phenomenon requiring explanation: the network is not merely memorizing the training samples, but appears to recover structure extending across an unseen orbit. More broadly, this provides a clean setting for studying how models exploit group-structured data and learn from far fewer samples than the ambient combinatorics would naively suggest. We agree that this motivation should be better signposted in the introduction.
>
> - *The community interest is never clearly stated.* We agree and will clarify this in the introduction. The intended audience is twofold: first, theoretical ML researchers interested in implicit bias, symmetry, and generalization, for whom Hopfield networks provide a minimal but analyzable setting; and second, researchers interested in Hopfield networks and associative memory systems themselves.
>
> - *The system under study (classical HNs) is a toy model and not linked to the modern architectures.* We respectfully disagree with characterizing classical Hopfield networks as merely a toy model: they are a foundational model of associative memory with a long history of theoretical study. That said, we agree that our work is not primarily motivated by immediate relevance to current large-scale architectures. One benefit of classical HNs is that the mechanism of interest can be exposed cleanly: fixed-point memorization becomes energy-gap inequalities; these inequalities become linear margin constraints in the Hopfield parameters; MEF becomes an exponential-loss problem over the induced constraints; and the resulting max-margin bias can be related to invariant parameter structure. This mechanistic clarity is a strength of the model for studying the emergence of invariance from group-structured data.
>
> - *What is learned that would not already be expected from combining Hillar et al. 2021 and Soudry et al. 2018?* The contribution is not simply the direct combination of these works. Soudry et al. explain the implicit bias of gradient descent toward max-margin solutions once a problem has already been formulated as separable exponential/logistic linear classification. A first nontrivial step in our work is to show that strict memorization in a classical Hopfield network can be written as a linear feasibility/margin problem in the Hopfield parameters, and that MEF corresponds to an exponential-loss problem over these induced constraints. Moreover, Soudry et al. do not address whether this max-margin bias leads to good generalization in a structured setting, nor how such a bias might be connected to emergent invariance. Hillar et al. identify the critical few-shot clique-learning phenomenon, observe important clique-symmetric parameter structure, and extend the setting to hypergraphs. However, they do not formulate Hopfield memorization as an induced margin/HSVM problem, do not analyze MEF through max-margin implicit bias, and leave the theoretical study of sample complexity / verification of the critical sample-ratio phenomenon as future work. We make progress on this through our sample-complexity bound, and our analysis applies to arbitrary graph isomorphism classes, not only cliques.
>
> Based on the reviewer’s comments, we will make the following changes. First, we will add one to two paragraphs in the introduction before stating the contributions, summarizing the research question, why the phenomenon is nontrivial, why classical HNs are an appropriate setting, who the intended audience is, and what is added beyond Hillar et al. and Soudry et al. Second, we will recalibrate the abstract, replacing “a unifying mechanism for generalization in Hopfield networks” with a statement explicitly restricted to orbit generalization in the graph-isomorphism setting. Third, we will add an explicit limitation stating that the immediate relevance of our results to modern architectures is not clear, and that extending these ideas to state-of-the-art models remains future work.